# Observations of Fragmented Aurora-like Emissions and Picket Fence on the Poleward Edge of the Auroral Oval

Sota Nanjo[1], Katie Herlingshaw[2], Tima Sergienko[1], Gaël Cessateur[5], Noora Partamies[2], Magnar G. Johnsen[3], Keisuke Hosokawa[4], Hervé Lamy[5], Yasunobu Ogawa[6], Antti Kero[7], Shin-ichiro Oyama[6,8,9], and Masatoshi Yamauchi[1]

[1]Swedish Institute of Space Physics (IRF), Kiruna, Sweden
[2]University Centre in Svalbard, Longyearbyen, Norway
[3]Tromsø Geophysical Observatory, UiT The Arctic University of Norway, Tromsø, Norway
[4]Graduate School of Informatics and Engineering, The University of Electro-Communications, Chofu, Japan
[5]Royal Belgian Institute for Space Aeronomy, Brussels, Belgium
[6]National Institute of Polar Research, Tachikawa, Japan
[7]Sodankylä Geophysical Observatory, Sodankylä, Finland
[8]Nagoya University, Nagoya, Japan
[9]University of Oulu, Oulu, Finland

**Correspondence:** Sota Nanjo (sota.nanjo@irf.se)

**Abstract.** We analyzed fragmented auroral-like emissions (FAEs) and picket fence structures observed in northern Scandinavia during a magnetic storm on January 1, 2025. The analysis is based on ground-based high-sensitivity optical observations and in-situ measurements from the Swarm satellites. While FAEs and picket fences have previously been reported in the polar cap and subauroral region, respectively, this study reports simultaneous occurrences of both phenomena in auroral latitudes, near the poleward edge of the oval. Ground-based camera observations revealed that some FAEs exhibited orientations closely aligned with the modeled local magnetic field in the image plane and appeared simultaneously at multiple longitudinally separated locations. Furthermore, the FAEs appeared to follow the motion of red auroras, suggesting that the background electric field structure and spatial gradients in the electron density may influence their formation. Consistent with previous studies, the generation of FAEs is considered to be due to local acceleration of electrons in the ionosphere rather than electron precipitation from the magnetosphere. While we could not clearly identify the generation mechanisms, the morphological diversity observed in this event suggests that multiple plasma instabilities may be involved in the generation of both FAEs and picket fence structures.

## 1 Introduction

Fragmented auroral-like emissions (FAEs) are a newly recognized optical phenomenon, primarily observed in the polar cap regions (Dreyer et al., 2021; Whiter et al., 2021; Partamies et al., 2025; Herlingshaw et al., 2025). These emissions are characterized by small scale and short duration. Like regular auroras, FAEs exhibit strong green emissions from atomic oxygen and first positive emissions from nitrogen molecules. However, unlike typical auroras driven by magnetospheric dynamics, FAEs do not show structures aligned with the magnetic field, suggesting that they may be a different phenomenon. Simultaneous

observations with high-speed imaging and incoherent scatter radar have indicated that FAEs may be related to waves generated by plasma instability in the ionospheric E region (Dreyer et al., 2021; Whiter et al., 2021).

Another type of aurora with morphological characteristics similar to FAEs is the "picket fence." This aurora is observed in the subauroral region and appears in association with Strong Thermal Emission Velocity Enhancement (STEVE; MacDonald et al. 2018; Nishimura et al. 2023). The name "picket fence" comes from its appearance: narrow, green rays aligned with the local magnetic field appear side-by-side along longitude, resembling a wooden fence. Because it occurs in conjunction with STEVE, the picket fence is also regarded as a subauroral phenomenon. Unlike FAEs, Semeter et al. (2020) suggested that the picket fence is field-aligned. Since FAEs and picket fence emissions have been observed at different latitudes, their relationship has not been systematically investigated. However, there are some similarities, such as their vivid green color in photographs taken by commercial digital cameras, short duration, and fine structure, suggesting that there may be some commonalities in their formation mechanisms. It has been suggested that electron precipitation from the magnetosphere plays a role in the formation of the picket fence (Nishimura et al., 2019; Mishin and Streltsov, 2019; Gillies et al., 2019), while others suggest that localized heating in the ionosphere is crucial (Mende et al., 2019; Semeter et al., 2020).

STEVE is observed in the subauroral region alongside high-speed plasma flows (MacDonald et al., 2018). It is also known to have a continuous spectrum (Gillies et al., 2019; Liang et al., 2019; Gillies et al., 2023), distinguishing it from typical auroras, which are characterized by discrete emission line spectra. Previous studies have suggested that the emission of STEVE may be related to high-speed ion flows caused by subauroral ion drifts (SAIDs; Chu et al. 2019; Nishimura et al. 2019; Archer et al. 2019; Martinis et al. 2021). However, the exact mechanism behind its optical signature remains a topic of debate (Nishimura et al., 2023; Liang and Donovan, 2024).

STEVE is defined to occur in the sub-auroral ionosphere in a longitudinally extended east-west arc that propagates in a westward direction. Emissions similar to STEVE have also been observed in the auroral region, and, like STEVE, high-speed ion flows were simultaneously measured (Nanjo et al., 2024). Additionally, continuum emissions have been observed in the polar cap boundary region (Partamies et al., 2025) and in auroral latitudes (Spanswick et al., 2024). Observations by Partamies et al. (2025) showed that the dayside continuum emission, whose spectrum is dominated by the red line, can be accompanied by a green aurora at lower altitudes. These continuum emissions show slight intensifications in wavelength regions where auroral emission lines are absent, and like STEVE, they may exhibit a continuous spectrum. Interestingly, continuum emissions are sometimes accompanied by FAEs. FAEs are thought to arise from local acceleration of electrons in the ionosphere rather than from particle precipitation from the magnetosphere. Considering that both continuum emissions and STEVE share a continuous spectrum, the relationship between them seems similar to that between STEVE and picket fence emissions. Therefore, by focusing on their commonalities, new insights into the formation mechanisms of FAEs and picket fence emissions may be obtained. However, since these phenomena occur at different latitudes, there have been few studies focusing on their relationship. In this study, we report the observation of FAEs and picket fence structures in auroral latitudes, likely near the poleward boundary of the auroral oval, expanding the known occurrence region of these phenomena beyond the polar cap and subauroral zones. By analyzing their simultaneous appearance from the same ground-based site, we examine their morphological characteristics and discuss possible generation mechanisms, including their relationship to each other.

## 2 Instruments

### 2.1 AI-feedback color all-sky camera

A color all-sky camera is installed at Skibotn Observatory in Skibotn, Norway (69.348° N, 20.363° E). This camera is a commercially available Sony $\alpha$6400, equipped with a Meike MK-6.5mm F2.0 fisheye lens. The exposure time is set to 8 seconds, and the ISO sensitivity is 5000. The acquired images have a resolution of 4000 × 4000 pixels and are saved in both JPEG and RAW formats. Observations are conducted when the solar elevation angle is below $-13°$. The default temporal resolution is 1 minute; however, it is shortened to 15 seconds when auroras are detected in the most recent image. The presence of auroras is evaluated using a deep learning model employed by Tromsø AI (Nanjo et al., 2022). During the night of January 1–2, 2025, the temporal resolution was 15 seconds for most of the time from the end of twilight until around 1:00 UT. The real-time observation images and classification results are available on the website (https://tromsoe-ai.cei.uec.ac.jp). Hereafter, this camera is referred to as the "AI camera."

### 2.2 Magnetometer

Since there is no magnetometer installed in Skibotn, we used data from the nearest geomagnetic station in Kilpisjärvi, Finland (69.06°N, 20.77°E), located 40 km southeast of Skibotn. This magnetometer is part of the IMAGE network (Tanskanen, 2009). The temporal resolution is 1 minute, and the horizontal component was used to evaluate geomagnetic disturbances and identify substorms.

### 2.3 Spectral riometer

A spectral riometer measures the absorption of cosmic radio noise over multiple frequency bands in the range of 10–80 MHz, allowing the estimation of electron precipitation into the D and E regions of the ionosphere. Stronger absorption indicates a greater amount of precipitating electrons (Kero et al., 2014). In this study, we used the spectral riometer installed in Kilpisjärvi, Finland (69.06° N, 20.77° E).

### 2.4 Auroral Spectrograph In Skibotn (ASIS)

The ASIS instrument, installed next to the AI camera, performs spectroscopic measurements of auroral emissions in the visible range (400–680 nm) at the magnetic zenith. The system consists of a guiding lens, an optical fiber, and a spectrograph with a CCD camera. The angle of view is approximately 4.5°, centered at an elevation of 77.26° and an azimuth of 182.79°, which corresponds to the magnetic field-aligned direction at the site. Temporal resolution is 30 seconds, and wavelength resolution is 0.3 nm. The guiding lens has a 60 mm diameter and a focal length of 248 mm, and directs light into an optical fiber via a reflective collimator. The spectrograph, an SR303-i Czerny–Turner model from ANDOR, includes an adjustable slit and selectable gratings (300, 600, 1800 lines/mm). A 16-bit iDUS CCD camera, cooled to -70°C, is used for detection with a resolution of 1024 × 256 pixels.

## 2.5 Watec all-sky cameras

Narrowband all-sky cameras are installed at Skibotn Observatory. The cameras are Watec WAT-910HX/RC, equipped with a Fujinon Fish-eye lens YV2.2x1.4A-SA2. Several Watec cameras are installed, each equipped with a different interference filter. The filters have a full width at half maximum (FWHM) of 10 nm, with central wavelengths of 430, 560, and 632 nm. The exposure time is 2 seconds for the 430 nm and 632 nm filters, and 1 second for the 560 nm filter. Since the Watec all-sky cameras acquire images continuously at fixed intervals, the temporal resolutions are 2 seconds and 1 second, respectively. The image size is 640 × 480 pixels. Through optical calibration, these cameras are capable of measuring absolute brightness in Rayleighs (Ogawa et al., 2020). The calibration parameters are provided in Appendix A. Hereafter, these cameras are referred to simply as "Watec cameras," and the wavelengths are represented as 428 nm, 558 nm, and 630 nm instead of the central wavelengths mentioned above.

## 2.6 qCMOS wide-angle camera

A high-speed camera is installed at Skibotn Observatory. The camera is the ORCA-Quest qCMOS (quantitative CMOS) camera from Hamamatsu Photonics, equipped with a Kowa Lens LM8HC F1.4 f8mm. The field of view (FoV) is 76° in the horizontal direction. After applying 4×4 hardware binning, images with a resolution of 1024 × 576 pixels are obtained. A BG3 glass filter is placed in front of the lens to block the green and red lines while maintaining sensitivity to the first negative and first positive bands of nitrogen molecule ions and molecules. Under normal operation, the camera captures 20 images per second; however, since this case was during a test operation, only one image per second was stored. The exposure time is 1/20 second.

## 2.7 Swarm satellites

Swarm satellites are polar-orbiting low Earth orbit (LEO) satellites (Olsen et al., 2013). This study uses data from the Electric Field Instrument (EFI) onboard Swarm A and C. The EFI includes a pair of Langmuir probes that provide measurements of spacecraft potential, electron density, and electron temperature with a temporal resolution of 0.5 s. Swarm A and C fly in close formation, enabling the estimation of field-aligned currents (FACs) with a temporal resolution of 1 s using the dual-satellite method. This study utilizes the Level 2 FAC-dual product, which improves accuracy by reducing assumptions about current sheet structures, providing reliable FAC estimates at high latitudes.

## 3 Observation

On January 1, 2025, intense auroral activity was observed over northern Scandinavia due to a magnetic storm that had begun the previous day. A notable characteristic of this event was the dominance of red auroral emissions. Additionally, multiple occurrences of FAEs and picket fence aurora were detected on this night. This study reports a simultaneous occurrence of both phenomena within the auroral zone on the same day. In this study, we focus on observational data obtained from multiple

optical instruments installed in Skibotn, Norway, to analyze the morphological characteristics of these auroras and consider the possible mechanisms behind their generation.

Figure 1 shows the time-series data of solar wind parameters and the Dst index associated with the magnetic storm that began on December 31, 2024, analyzed in this study. The solar wind data were measured over three days from December 31, 2024, to January 2, 2025, by the DSCOVR satellite located close to the L1 point. From top to bottom, the panels represent the interplanetary magnetic field (IMF), dynamic pressure, proton speed, proton density, and the real-time Dst index. Around 15:45 UT on December 31, 2024, the arrival of a coronal mass ejection (CME) that occurred on December 29 caused a sudden

increase in IMF intensity, dynamic pressure, speed, and density, leading to a storm sudden commencement (SSC).

    After the SSC arrival, the IMF north-south (z) component fluctuated, but a few hours later, it settled between $-5$ nT and $-10$ nT for about 10 hours. Since the solar wind speed was moderate, around 450 km/s, the magnetic storm did not develop significantly (the minimum real-time Dst index until 09:00 UT on January 1 remained above $-50$ nT). Subsequently, the IMF z component gradually intensified, falling below $-20$ nT. The increase in dynamic pressure also contributed to the development of the main phase of the storm, causing the real-time Dst index to drop below $-200$ nT.

Between 16:10 UT and 17:40 UT on January 1, DSCOVR recorded a sudden increase in solar wind density. The peak solar wind density reached 101 /cc, which was higher than that observed during the May 2024 magnetic storm (e.g., Spogli et al., 2024; Tulasi Ram et al., 2024). At the same time, the southward IMF turned northward. After this time, the IMF did not become significantly southward again, and the magnetic storm entered the recovery phase.

Intense magnetic disturbances were observed on the ground as well. Figure 2 presents the ground-based observations in northern Scandinavia during the night of January 1–2. The panels show (a) the horizontal (H) component of the ground magnetic field, (b) cosmic noise absorption measured by a spectral riometer, (c) a keogram from the AI camera, and (d) auroral brightness variations at the magnetic zenith measured by the spectrograph. Panels (a) and (b) were recorded in Kilpisjärvi, Finland, while panels (c) and (d) were obtained in Skibotn, Norway. Kilpisjärvi is located approximately 40 km southeast of

Skibotn. Given how close they are, Kilpisjärvi and Skibotn share nearly the same magnetic latitude (MLAT) and magnetic local time (MLT). According to Figure 2(a), two substorms occurred during the night, around 17:40 UT and 21:00 UT. The earlier event was particularly strong, with the H component decreasing by approximately 2000 nT, making it an exceptionally intense substorm.

    During the first substorm, strong cosmic noise absorption was observed in Figure 2(b), suggesting significant electron pre-

cipitation into the D and E regions of the ionosphere. Correspondingly, Figure 2(c) shows a bright poleward expansion with a colorful display of white, green, and red emissions. Around 21:00 UT, a northward expansion of substorm onset was also observed; however, its brightness was weaker compared to the first substorm.

    According to the spectrograph measurements in Figure 2(d), during the first substorm, the brightness at 557.7 nm reached a maximum of 275 kR, while the combined brightness at 630.0 nm and 636.4 nm peaked at 216 kR. The brightness at 427.8 nm,

associated with the first negative system of molecular nitrogen ions, peaked at 45 kR. An increase in auroral brightness was also observed during the second substorm; however, the intensity was an order of magnitude lower compared to the first substorm. Additionally, red emissions were stronger than the green emissions, even though the latter are typically the brightest.

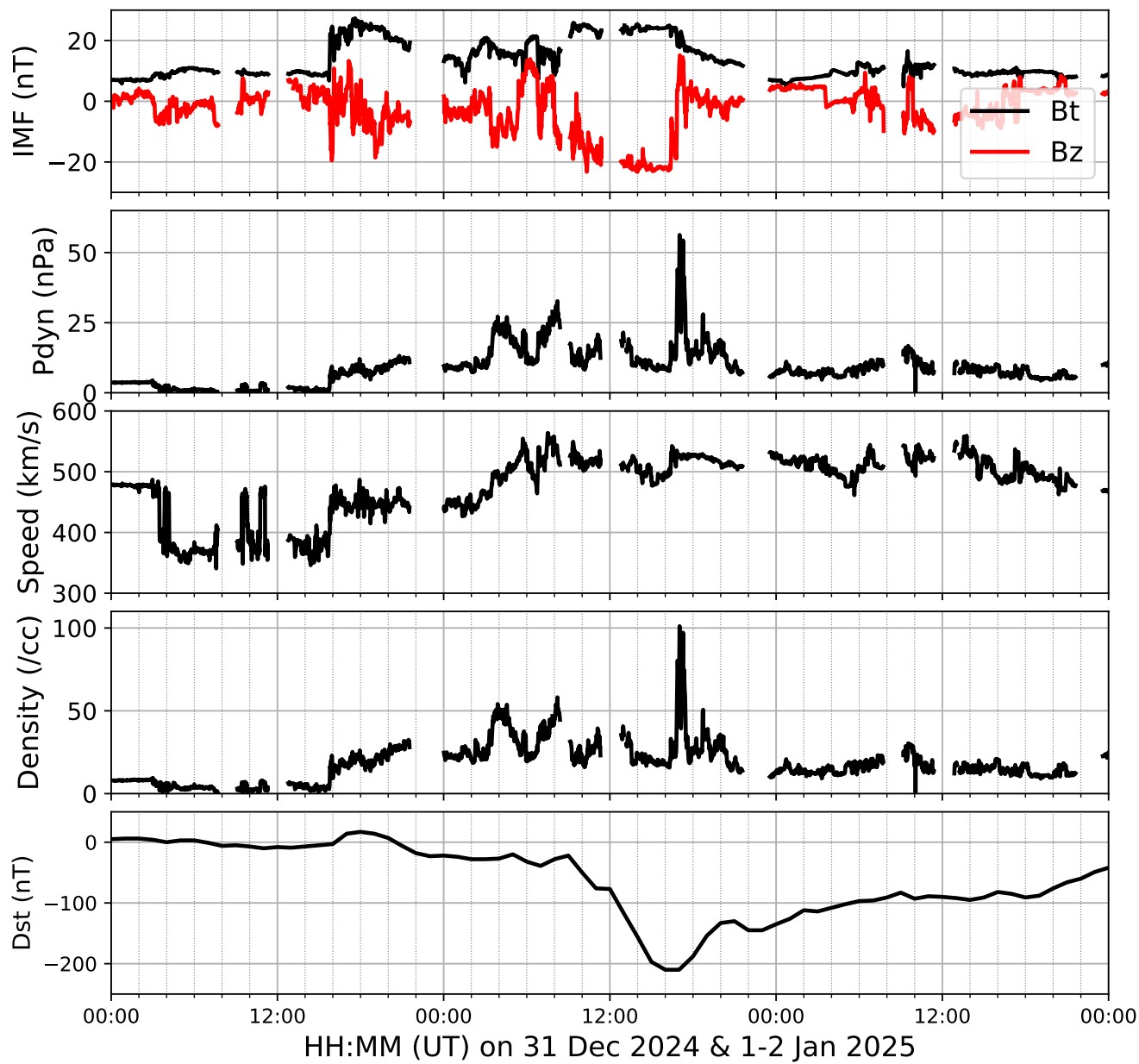

**Figure 1.** Solar wind parameters measured by DSCOVR near the L1 point and the Dst index. From top to bottom, the panels show the interplanetary magnetic field (IMF), dynamic pressure, proton speed, proton density, and the real-time Dst index. The data cover the period from December 31, 2024, to January 2, 2025. The sudden increase in IMF intensity, solar wind pressure, and proton speed at around 15:45 UT on December 31 marks the arrival of a coronal mass ejection (CME), leading to the onset of the storm.

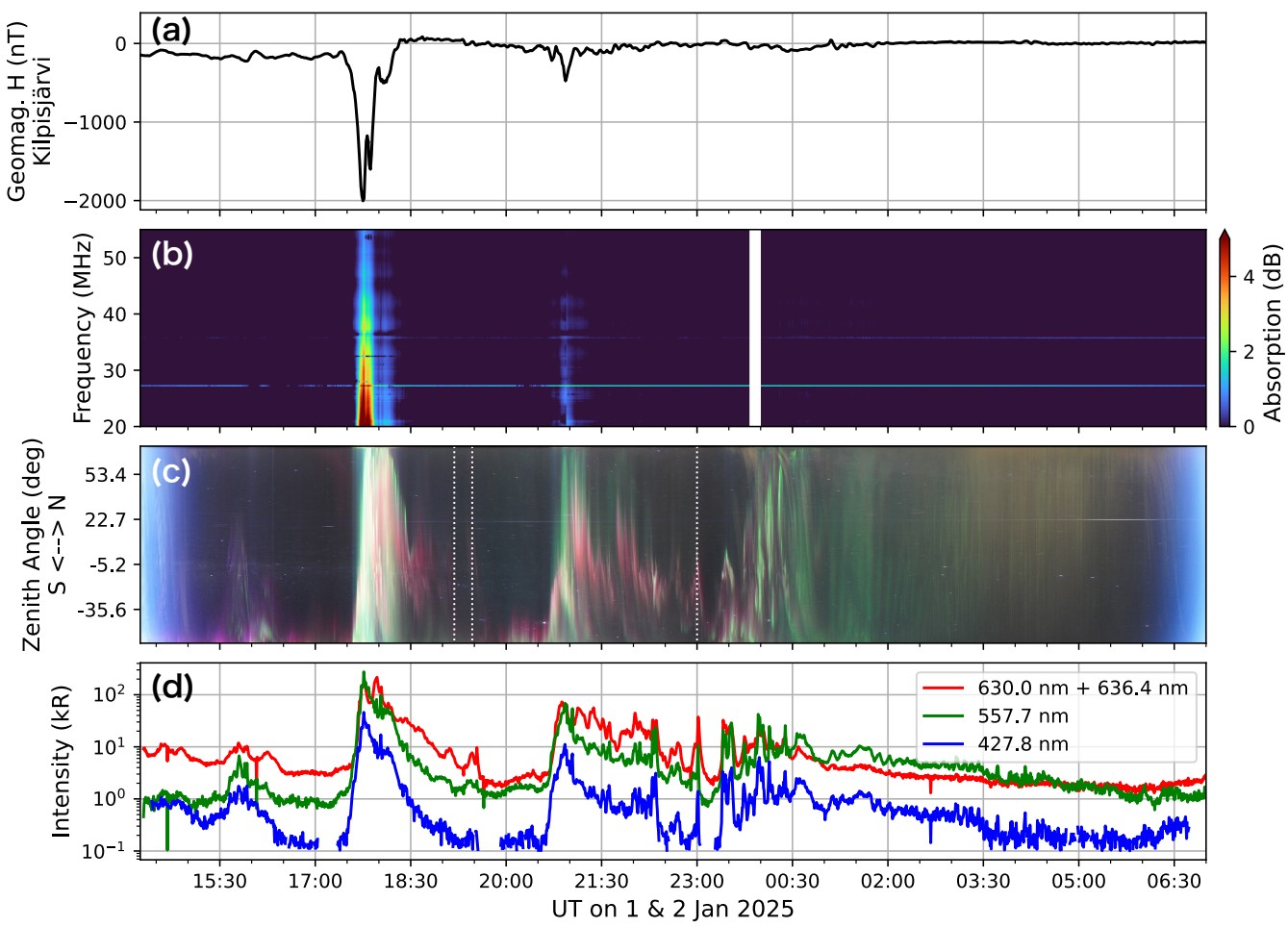

**Figure 2.** Ground-based observations in Kilpisjärvi and Skibotn during the night of January 1–2, 2025. The panels show (a) the horizontal (H) component of the ground magnetic field, (b) cosmic noise absorption observed by the spectral riometer, suggesting precipitation of high-energy electrons, and (c) and (d) auroral emissions measured by the all-sky camera and spectrograph, respectively. A particularly strong disturbance occurred around 17:40 UT, as seen in panel (a), which corresponds to enhanced cosmic noise absorption in panel (b).

This indicates that the precipitation of low-energy electrons was enhanced compared to typical conditions. After midnight, the green line became dominant, and normal green auroras were observed in Figure 2(c).

FAEs and picket fences were observed during the night when solar wind density increased sharply, and red aurora was dominant. The times of these events are marked by white vertical dotted lines in Figure 2(c). These occurrences were at 19:11, 19:28, and 23:00 UT, which we refer to as Event 1, Event 2, and Event 3, respectively. As seen in Figures 1(a) and 1(c), all of these events occurred outside the periods of poleward expansions associated with substorms. During Events 1 and 2, auroral brightness was relatively weak, with red emissions of 7–8 kR and green emissions of 1–2 kR. In contrast, Event 3 exhibited

higher auroral brightness, with red emissions ranging from 10–40 kR and green emissions from 3 to 10 kR. Although there was a significant difference in brightness among these events, a common feature was that red emissions were the strongest.

As shown in Figure 3, just several minutes before Event 1, a red-dominated aurora was observed near the zenith of Skibotn, exhibiting vortical motion. The video version is provided as Video A1. As can be seen in the video, the red aurora shows no significant morphological evolution between 19:08:33 and 19:11:06 UT; therefore, frames between Figures 3 and 4 are not

shown. The red aurora propagated westward on the low-latitude (south) side and eastward on the high-latitude (north) side, consistent with the direction of the $\mathbf{E} \times \mathbf{B}$ drift if we assume that Pedersen currents close the upward FAC in the red arc. The white arrows do not indicate the direction of motion itself but rather mark auroral regions exhibiting the most apparent temporal changes. The east-west motion of the red aurora can be tracked by following the sequence of panels. Assuming an emission height of 200–400 km, the speed is estimated to be 2.3–4.6 km/s. Additionally, the auroral brightness at the magnetic zenith

during this time was 200 R at 427.8 nm, 2 kR at 557.7 nm, and approximately 8 kR for the combined emissions at 630.0 nm and 636.4 nm.

As shown in Figure 4, a few minutes after the shear flow of the red aurora was observed, FAEs were detected around 19:11 UT. They were located near the zenith, indicated by the white arrows, and appeared approximately 100 km poleward (northward) from the red aurora. While this distance strictly depends on the emission heights of the respective phenomena,

both occurred close to the zenith and thus the values are expected to be similar. The AACGM coordinates (Shepherd, 2014) of FAEs were MLAT of 66.8° and MLT of 20.8. Although the spectrograph does not provide spectra of FAEs due to its focus on the magnetic zenith, a Watec camera equipped with a narrowband filter showed that the FAEs were dominated by the green line emission, with no detectable emissions at 630 nm or 428 nm, or very weak emissions below the detection threshold (~115 R, see Appendix A). Additionally, during Event 1, as shown by the yellow and orange lines, the Swarm satellite passed through

the FoV of the AI camera, although they were approximately 500 km apart. As shown by the white rectangle, the FAEs were captured within the FoV of the qCMOS camera. This enabled the tracking of their movement at a 1-second temporal resolution.

Figure 5 shows snapshots of the qCMOS camera observations during Event 1. A video version is also provided as Video A2. Panel (a) shows the full image, with the region enclosed by the white square corresponding to panels (b) through (i). Magnetic field lines, calculated using the Tsyganenko 89 model (Tsyganenko, 1989), are depicted by yellow dots. The lowest

points (footpoints) of these model field lines were placed at an altitude of 110 km at geographic latitude 69.6° and longitudes 19.0°–21.0° in 0.2° steps. When we vary the starting height between 100 and 140 km, the location of the magnetic zenith and the relative alignment between the FAEs and the modeled field lines change little. Although some FAEs in individual

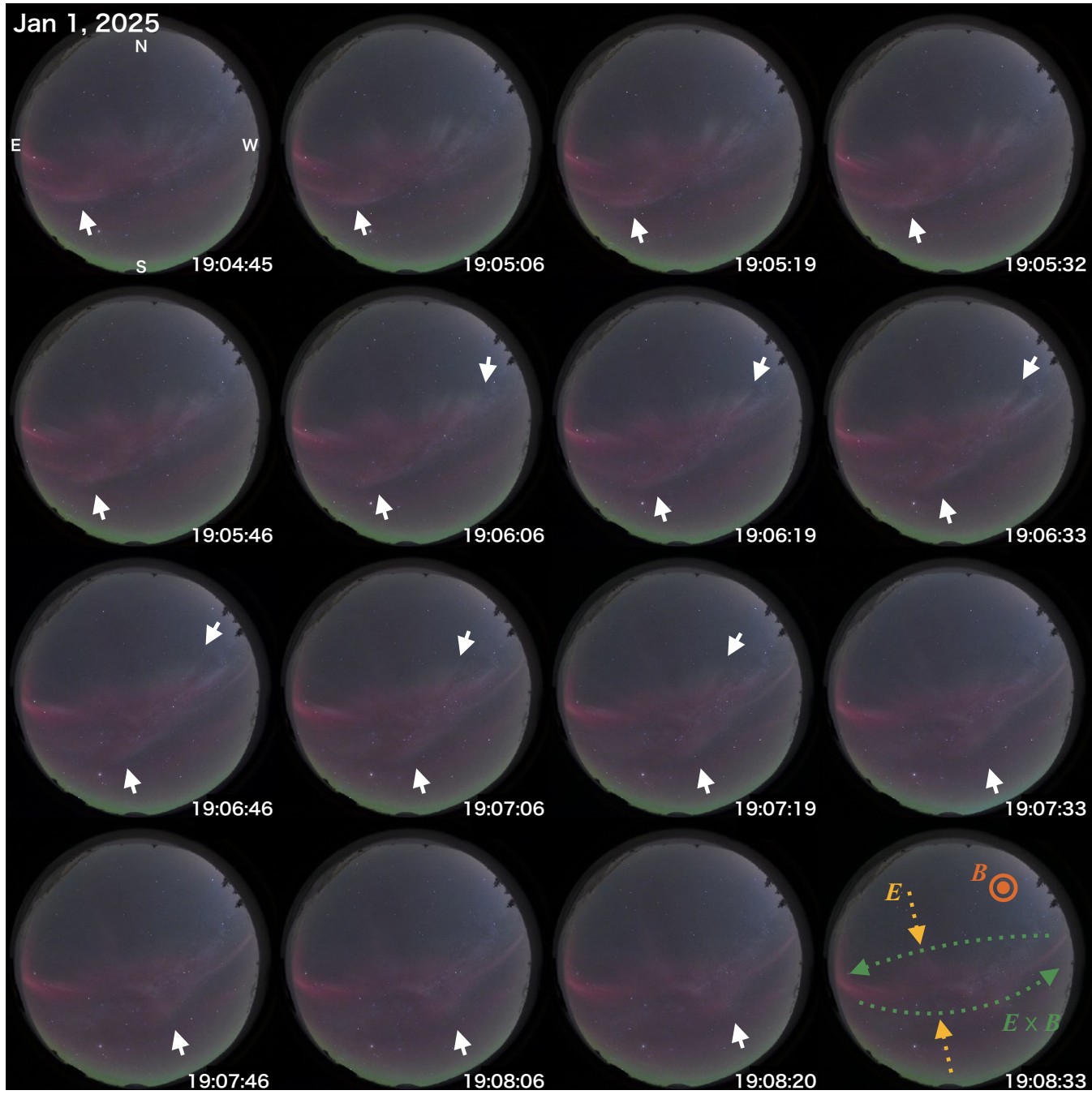

**Figure 3.** The shear flow of the red aurora observed before Event 1. The images show the spiral motion of the red aurora near the zenith of Skibotn. The video, provided as Video A1, further illustrates the dynamics of the auroral movement. The white arrows indicate the direction of propagation with westward motion on the low-latitude (south) side and eastward motion on the high-latitude (north) side.

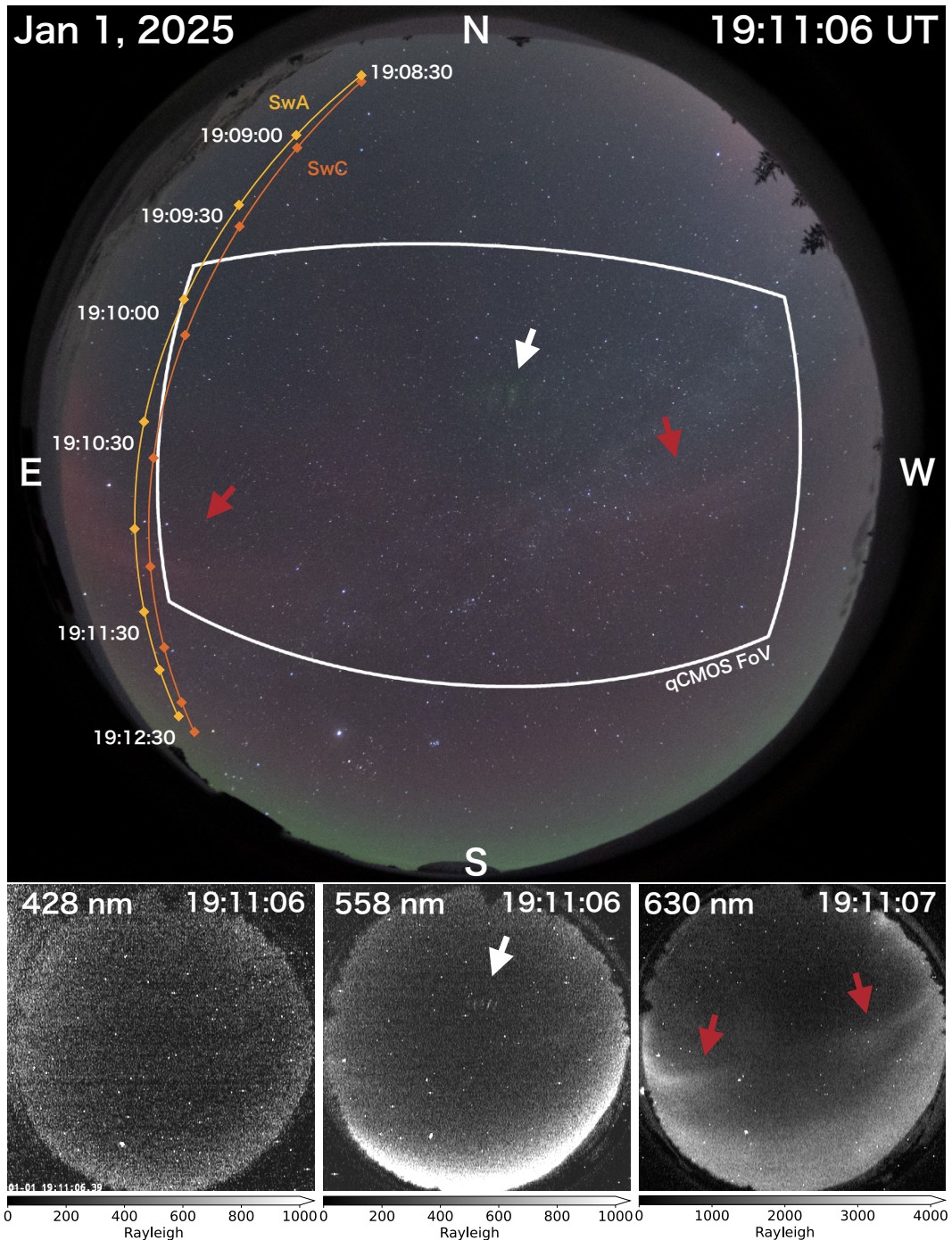

**Figure 4.** FAEs during Event 1. The top panel shows a color image from the AI camera, while the bottom panels display monochrome images from the Watec cameras equipped with narrowband filters. The top panel also includes the FoV of the qCMOS camera and the paths of the Swarm satellites. Based on the Watec camera data, the green line emission seemed to be the most prominent among the three wavelengths.

snapshots do not appear perfectly parallel to the model field lines, the sequence shows that their orientations track the local field-line inclination across the FoV. Furthermore, the detailed image analysis presented in Appendix A2 shows that the angular difference between the extracted FAE orientations and the modeled magnetic field lines is approximately $4°$ in the image plane. Taken together, these results are not inconsistent with the assumption that the FAEs are parallel to the magnetic field in three dimensions, while their true three-dimensional geometry cannot be uniquely determined from a single-site view. Each FAE appeared regularly with a spacing of about 5 km in the east-west direction, and the vertical extent was approximately 10 km if we assume the bottom altitude is 110 km. The FAEs propagated eastward at a speed of around 200 m/s, disappearing after 15–20 seconds. New FAEs emerged in the eastward direction, resulting in a total duration of about 30 seconds. The spatial scale and lifetime were consistent with previous studies (Dreyer et al., 2021). However, to our knowledge, the alignment of the structures with the magnetic field lines has not been reported before.

At the same time, the Swarm satellites flew over northern Scandinavia. The measurement results are shown in Figure 6. Swarm A/C followed an approximately north-to-south orbit, and as shown in Figure 4, they flew within the camera's FoV from 19:08:30 for about 4 minutes, although they did not cross the FAE directly. The panels in Figure 6, from top to bottom, show the eastward component of the magnetic field after removing the IGRF component, electron density, electron temperature, and field-aligned currents (FACs) derived using data from Swarm A/C. Referring to Figure 4, red aurora was observed around 19:11:00 UT. From Figures 6(a) and 6(b), it is evident that the measurements from Swarm A and C are nearly identical, with only several seconds of time shift. Therefore, the results in Figure 6 likely represent spatial variation rather than temporal variation.

Based on the FACs derived from Swarm magnetic field measurements, the upward FAC seen until around 19:11:30 (on the high-latitude side) corresponds to the region 1 FAC, with the downward region 2 FAC starting afterward. As seen in Figure 4, the interval around 19:11:00 UT, when Swarm A/C crossed the red aurora, coincides with the electron-density enhancement labeled "Red aurora?" in Figure 6. We therefore interpret this enhancement as corresponding to the red aurora and being associated with upward Region 1 FACs. Because the FAEs occurred in a very confined region that was offset from the Swarm trajectories, we do not attempt to associate the Swarm measurements with the FAEs themselves.

Enhancements in electron density were measured that may correspond to the FAEs and red aurora. As indicated by the black arrows, around 19:11:00 UT, the electron density in a certain region increased to nearly twice the previous plateau ($\sim$19:10:30 UT), reaching approximately $2 \times 10^5$ cm$^{-3}$. After this increase, the satellites went to the latitude of the red aurora, where the electron density further increased to about $3 \times 10^5$ cm$^{-3}$. In Figure 6(c), the electron temperature does not show a gradient similar to the density; however, the absolute value was high, around 3000 K. According to Kwagala et al. (2017, 2018), conditions with $T_e > 2300$ K and $N_e > 2 \times 10^5$ cm$^{-3}$ allow red emission from oxygen to be produced by local collisions of thermal electrons. Since these conditions were satisfied, the latter density peak likely corresponds to the observed red aurora. By contrast, because green (557.7 nm) emission requires higher excitation energy than the red line, even if a density gradient comparable to that in Figure 6(b) exists near the FAEs, it remains questionable whether such a gradient alone would be decisive for their formation.

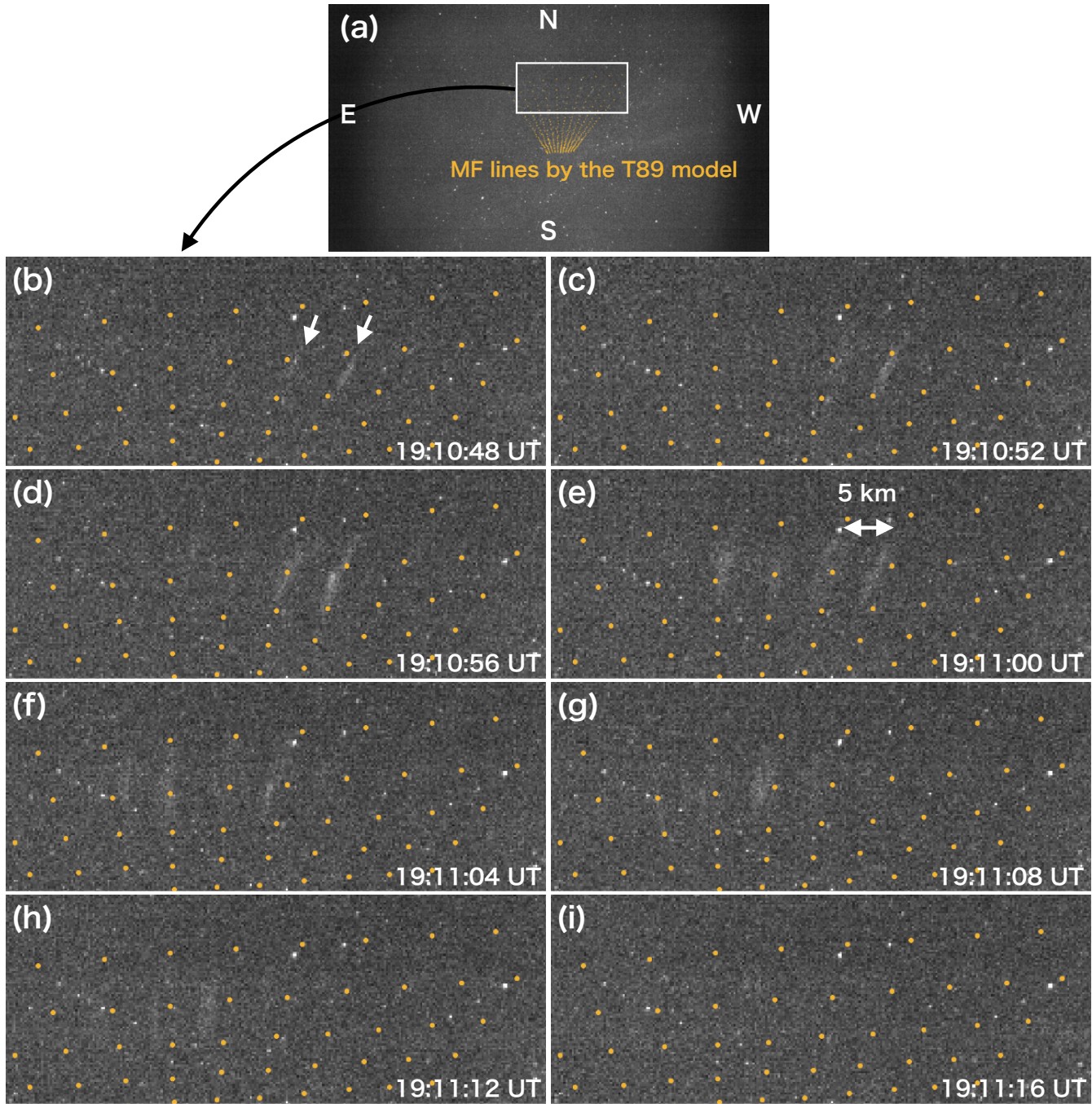

**Figure 5.** FAEs observed with the qCMOS camera during Event 1. The yellow dots represent magnetic field lines calculated using the Tsyganenko 89 (T89) model. Panel (a) shows the full image, and panels (b–i) show snapshots at 4 s cadence from 19:10:48 to 19:11:16 UT, illustrating that the overall group of FAEs lasts for roughly 30 s. The FAEs are consistent with alignment to the local modeled field lines.

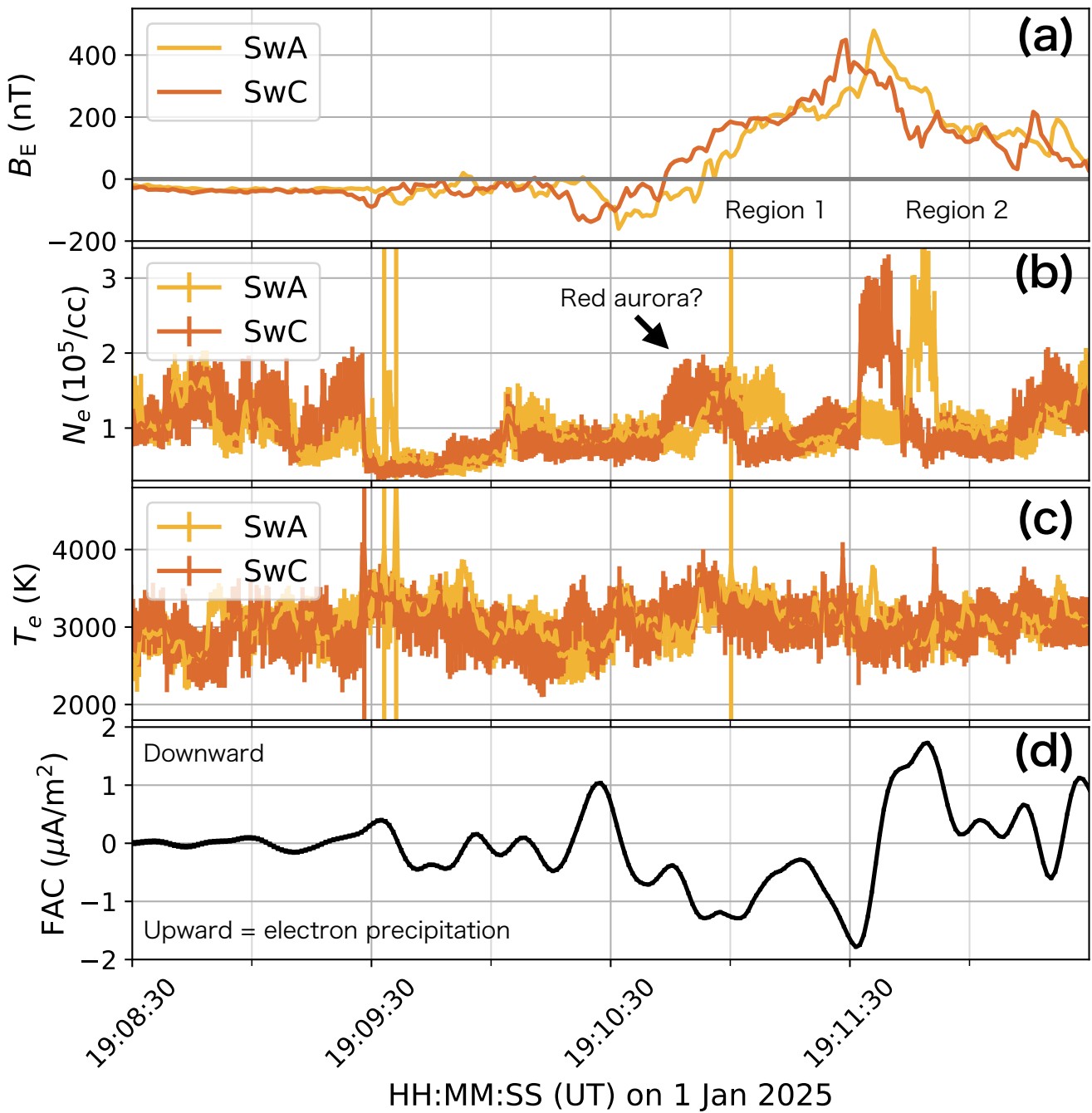

**Figure 6.** Measurement results from Swarm A/C. The panels, from top to bottom, show the eastward component of the magnetic field after subtracting the IGRF component, electron density, electron temperature, and field-aligned currents. The arrow labelled "Red aurora?" marks the enhancement that we interpret as corresponding to the red auroral arc shown in Figure 4. The areas labelled "Region 1" and "Region 2" in (a) indicate the corresponding FAC regions.

During Event 2, the picket fence and FAEs were observed in different locations. The picket fence appeared on the northern side of the FoV, and a cropped section of this region is shown in Figure 7. As indicated by the white arrows, the picket fence was first observed at 19:22:07 UT, initially appearing in the northwest direction and later propagating northeast. Additionally, as indicated by the red arrows, a red aurora appeared near the picket fence. This was consistent throughout the propagation of the picket fence, meaning that both the picket fence and the red aurora propagated in the same direction. The propagation speed of each was approximately 700–800 m/s. Furthermore, measurements from the Watec camera confirmed that the red aurora was associated with the 630.0 nm emission.

The picket fence initially appeared in a distinct characteristic shape, but as it propagated northeast, its structure seemed to fragment, becoming more similar to that of the FAEs. This change in appearance occurred when the FAEs crossed to the opposite side of the red aurora. However, the picket fence did not remain visible throughout the observation sequence and would occasionally disappear, so it is unclear whether the two shapes are the same phenomenon. Nevertheless, it is a fact that the picket fence and FAEs appeared at the same time and nearby.

During this period, snapshots from a webcam installed by the tourist company Lights Over Lapland in Abisko, Sweden, as shown in Figure 8(a), were also available. Since this is a business-use camera, the exact location of the camera installation cannot be disclosed. However, it was installed within a 1 km radius of the Abisko tourist station (68.357° N, 18.782° E) and faces northwest. This camera is used for real-time video streaming, but only snapshots with a 5-minute temporal resolution are recorded for archival purposes.

In Figure 8(a), a green picket fence appears from the center to the right side, with multiple red auroras observed around it. Using this image along with the image observed from Skibotn (Figure 7), the location of the picket fence was estimated. Figure 8(b) projects Figure 8(a) onto a longitude–altitude plane at a geographic latitude of 71.13°, while Figures 8(c) and 8(d) project images captured by the AI camera in Skibotn onto the same plane. We selected the latitude slice (at 71.13°) that makes the picket fence features from Abisko and Skibotn align most closely in the longitude–altitude projection. Although the images were taken at different times, they generally showed good agreement. According to this, the picket fence was distributed at altitudes between 110 and 140 km, and the longitudinal separation between adjacent pillars was approximately 5 km. In AACGM coordinates, MLAT was 68.9°, and MLT was 20.9.

In Event 2, FAEs were observed in addition to the picket fence. The summary is shown in Figure 9. As indicated by the white arrows, similar to Event 1, a shear flow of auroras in the longitudinal direction was observed. The FAEs appeared in two phases, with the first appearance around 19:25 UT. The aurora propagated from the northwest to the southeast at a speed of approximately 400–500 m/s, followed by the appearance of the FAEs, which tracked the aurora. This is illustrated in Figure 9(o). The aurora and FAEs maintained a distance of approximately 80 km. Later, around 19:30, the aurora moved further southeast, and additional FAEs appeared. This is depicted in Figure 9(p). At this time, the number of FAEs was higher compared to 19:25. The distance between the aurora and FAEs remained consistent, similar to the earlier observation. The FAEs were distributed around 67–68 MLAT and 20.2–21.0 MLT.

The FAEs that occurred around 19:30 UT in Event 2 were also observed with the qCMOS camera, and unlike Event 1, they were found to lack a structure aligned with the magnetic field lines. The results are shown in Figure 10. Similar to Figure 5,

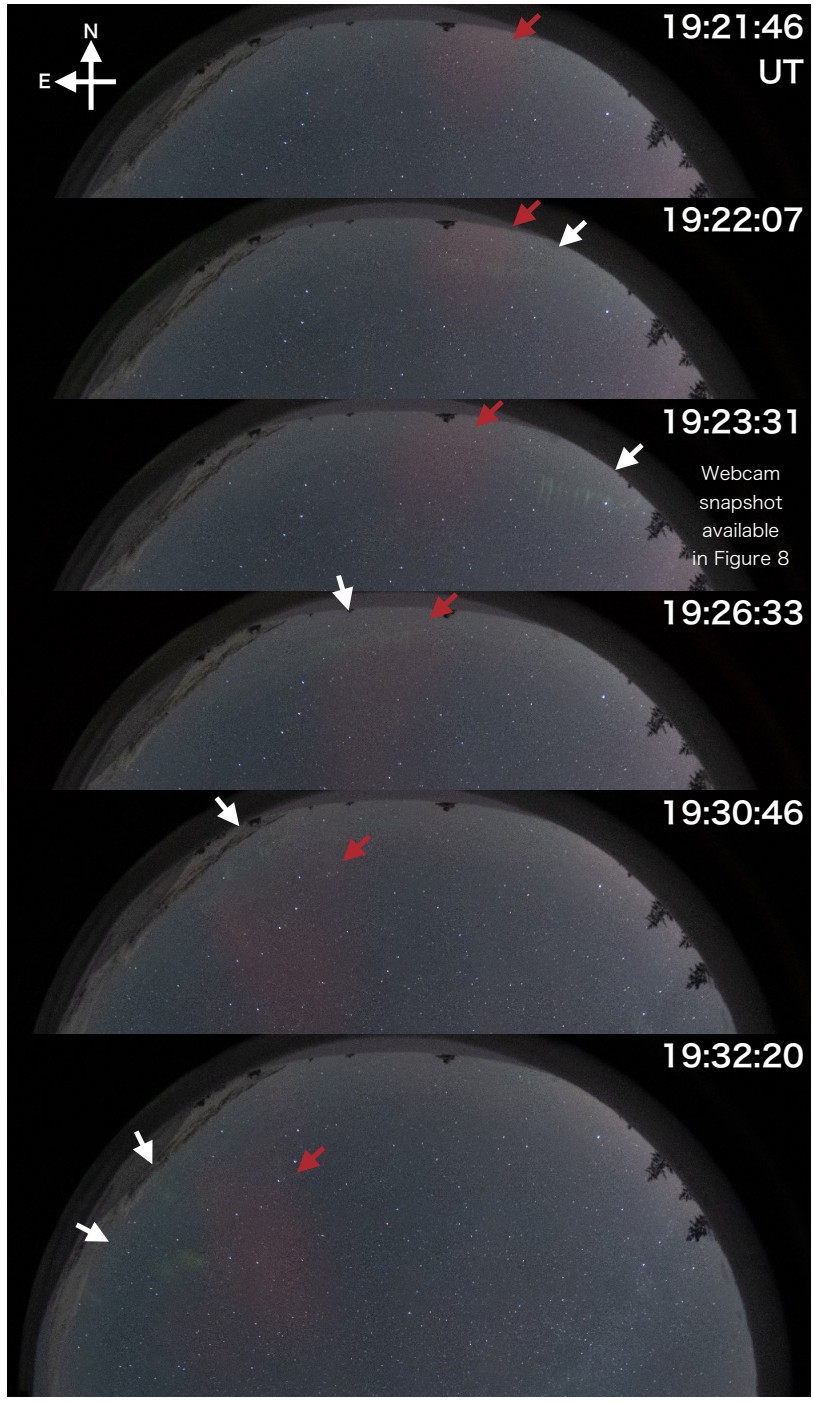

**Figure 7.** The propagation and morphological changes of the picket fence from its emergence to dissipation observed by the AI camera.

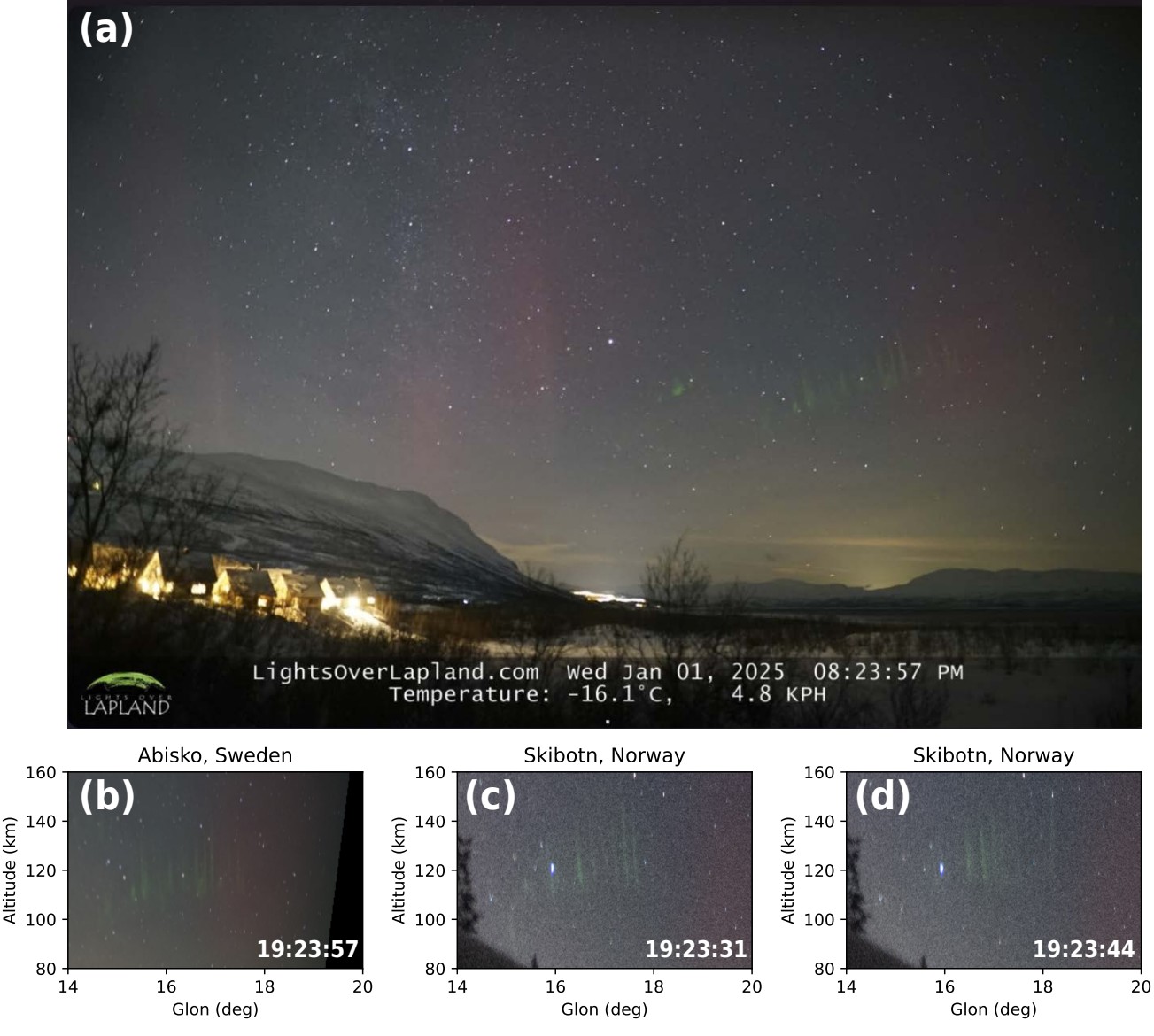

**Figure 8.** (a) An all-sky image showing a green picket fence structure extending from the center to the right side, accompanied by multiple red auroras. The photograph is reproduced with permission from Lights Over Lapland AB. (b) Projection of panel (a) onto a longitude–altitude plane at a geographic latitude of 71.13°. (c, d) Projections of images captured by the AI camera in Skibotn onto the same plane. Although the images were taken at different times, resulting in slight morphological differences, the overall structures show good agreement. The picket fence was located at altitudes between 110 and 140 km, and the longitudinal separation of each pillar was approximately 5 km.

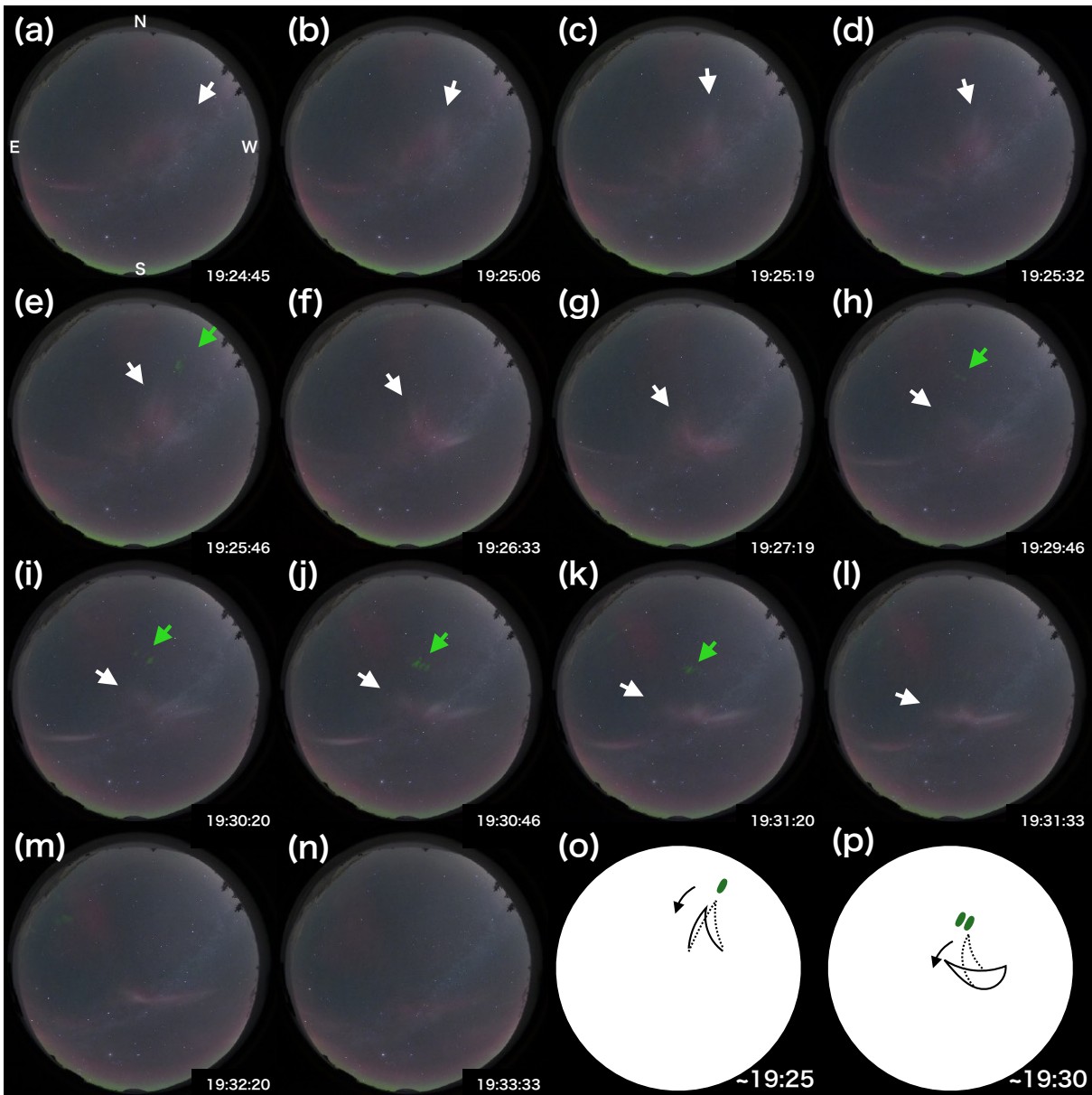

**Figure 9.** Summary of observations for Event 2. (a–n) Sequence of all-sky images showing the evolution of auroras and fragmented auroral-like emissions (FAEs). White arrows indicate the longitudinal shear flow of auroras, similar to that observed in Event 1, and green arrows point to the FAEs. (o) Illustration of the first phase of FAE appearance around 19:25 UT. The aurora propagated from northwest to southeast at a speed of approximately 400–500 m/s, followed by the appearance of FAEs that tracked the aurora while maintaining a distance of about 80 km. (p) Illustration of the second phase around 19:30 UT, where the aurora moved further southeast and additional FAEs appeared. The number of FAEs increased compared to the earlier phase, while the distance between the aurora and FAEs remained similar.

the region to the north of the zenith is cropped. A video version is available as Video A2. The FAEs were observed around 19:29:32 UT. The brightness was low initially, and the spatial scale was small, approximately 10 km in the longer directions. In the first-row panels, the white and orange arrows follow the movements of different FAEs, with each arrow consistently tracking the same FAE throughout. According to this, the FAEs propagated northeast at a speed of approximately 1 km/s. The yellow magnetic field lines, shown in Figure 5, were not parallel to the structure of the FAEs, indicating that the FAEs lacked a structure aligned with the magnetic field lines. The yellow arrow indicates the aurora, which appeared to maintain a distance from the FAEs, as seen in Figure 9.

The second row of Figure 10 shows images taken about one minute after the first row. Compared to the first row, the FAEs propagated to lower latitudes (southward). The same FAEs observed in the first row did not continue to appear; rather, the old FAE in the northeast disappeared, and a new FAE appeared in the southwest. This process was repeated, resulting in the FAEs overall propagating to lower latitudes. The aurora that appeared in the southern region also moved to lower latitudes, and as a result, the distance between the FAEs and the aurora remained nearly constant at 20–30 km. Additionally, when comparing the last two panels of the four panels in the second row, the FAEs initially appeared as small granular structures, which seemed to grow and extend southeastward. Since this is an observation from a single location, it is unclear whether this growth is truly southeastward or if it appears that way due to growth in the altitude direction.

As shown in the third row, one minute later, the number of FAEs observed simultaneously decreased to around two, and their brightness also diminished. During this period, the longitudinal movement of the aurora in the southern region was clearly observed, as indicated by the red arrows. The white arrows show the movement of the FAEs observed at the same time, indicating that both the aurora and FAEs propagated in the same direction.

In Event 3, FAEs were observed to the north (poleward) of the bright, southward-propagating discrete aurora. As shown in Figure 11(a), the FAEs appeared in several regions separated by approximately 500 km in the longitudinal direction (0.1–0.8 MLT), but all FAEs were located to the north (poleward) of the discrete aurora.

Figures 11(b), 11(c), and 11(d) show the observational results from the Watec camera, where the FAEs were detected only in the green line. The aurora, which is predominantly red and white in Figure 11(a), was brightest in the red line, as shown in Figure 11(d), and was observed in all three images. The spectrum of this aurora at the magnetic zenith is shown in Figure 11(e). According to this panel, the red emissions at 630.0 nm and 636.4 nm were the strongest. In the all-sky image taken by the AI camera, the aurora appears white; however, the spectrum shows that the contributions from the forbidden lines of atomic oxygen are two orders of magnitude stronger than those at other wavelengths, suggesting that the color seen in the images is primarily determined by these emissions. Therefore, the fact that the aurora appears whitish in the all-sky images does not necessarily indicate the presence of continuum emissions, meaning it cannot be determined from the image alone whether the observed aurora is a normal aurora or a phenomenon referred to as continuum or STEVE.

In Event 3, FAEs appeared near the zenith, and as shown in Figure 12, observations from the qCMOS camera were available. A video version is available as Video A2. The FAEs began to appear at 23:00:20 UT, and like in Event 2, they lacked a structure aligned with the magnetic field lines. Additionally, each FAE propagated eastward while the overall distribution shifted to lower latitudes (southward). The discrete aurora that appeared to the south of the FAEs also propagated southward, but the speed of

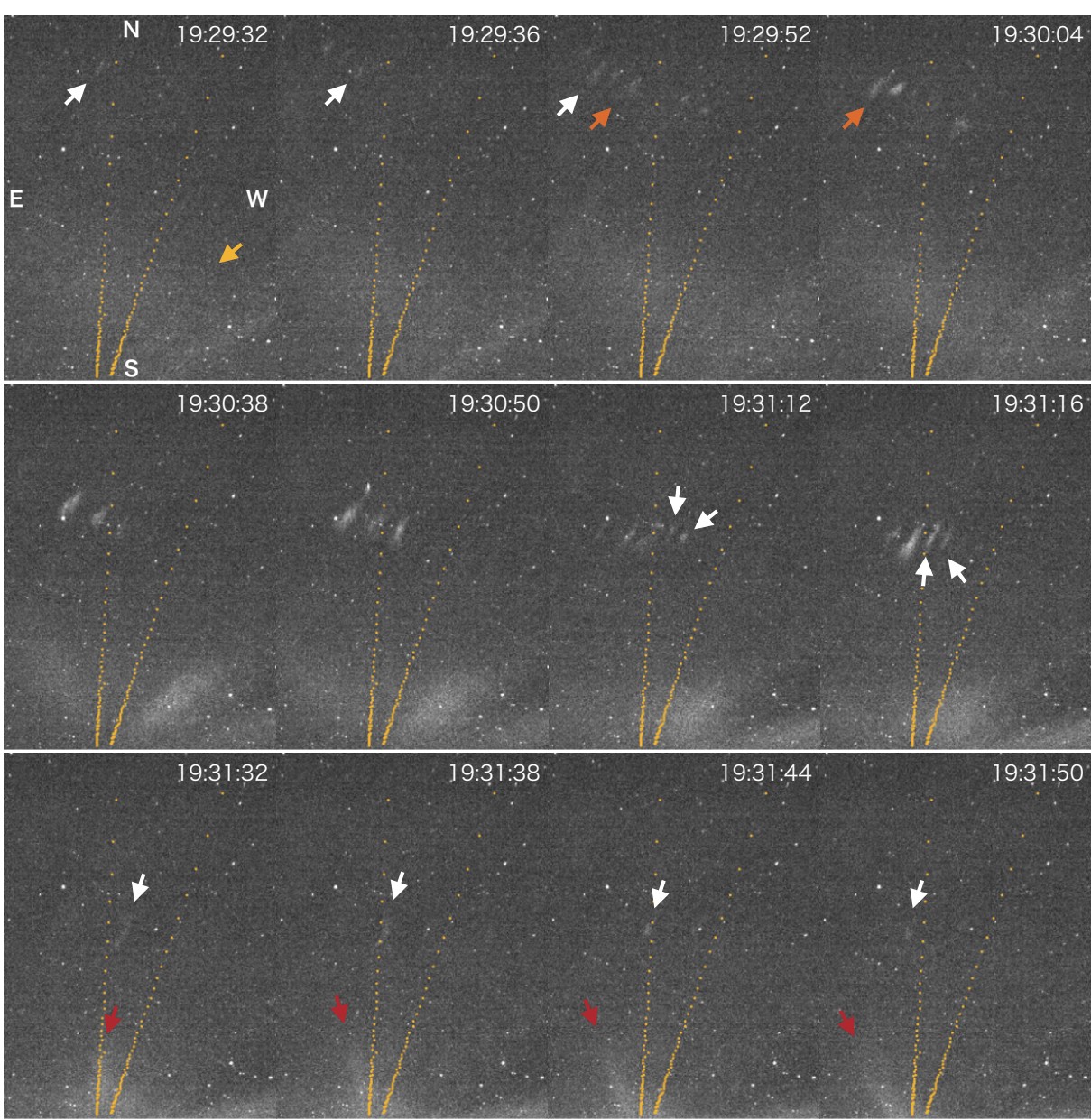

**Figure 10.** qCMOS camera observations of FAEs around 19:30 UT during Event 2. FAEs initially appeared around 19:29:32 UT, propagating northeast without field-aligned structure. About one minute later, new FAEs emerged to the southwest, resulting in a southward shift while maintaining distance from the aurora. Some FAEs appeared as small granular structures and seemed to grow southeastward. Later, the number and brightness of FAEs decreased, with both the aurora and FAEs moving in the same general direction.

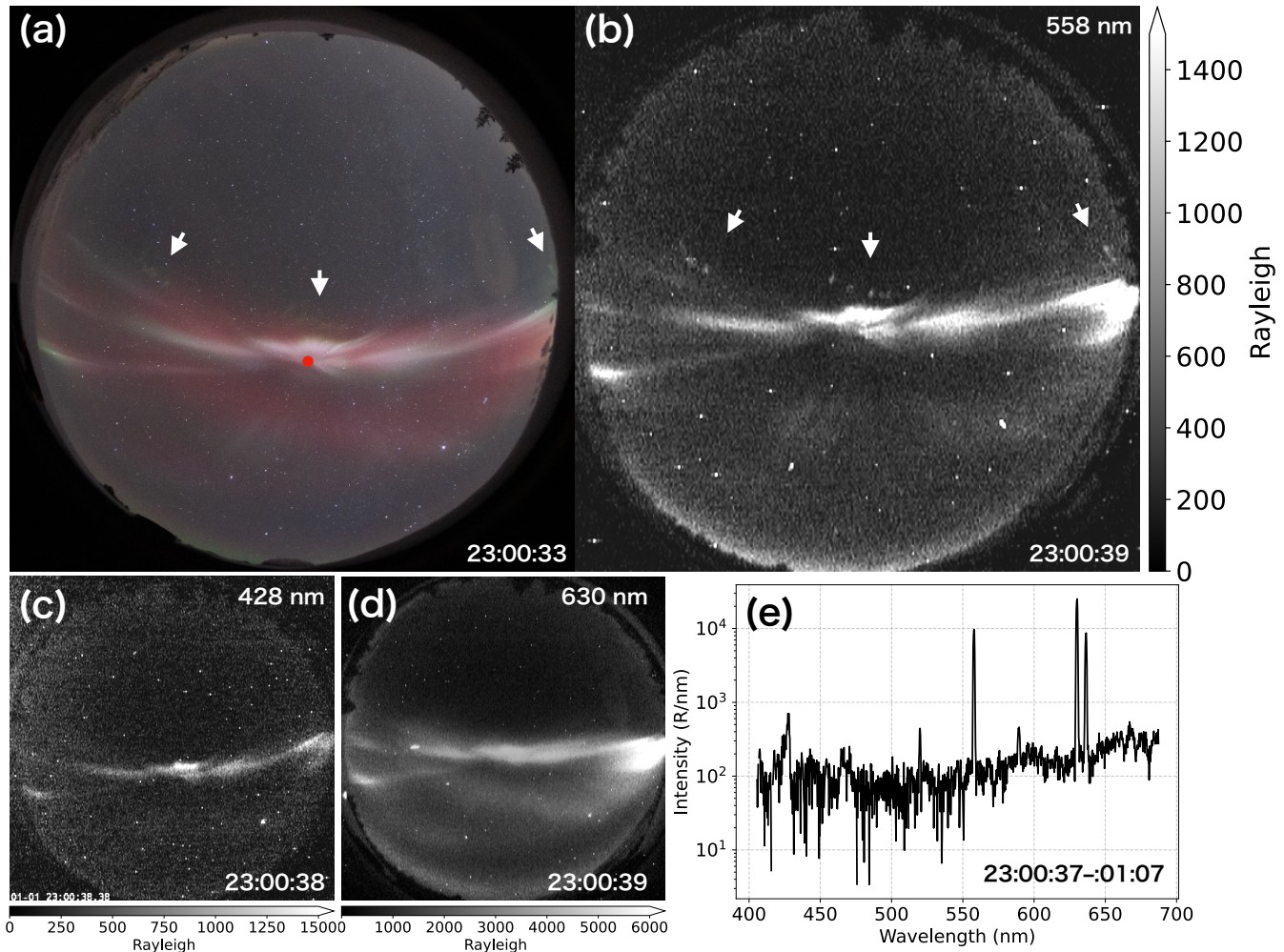

**Figure 11.** Observations of FAEs during Event 3. (a) All-sky image showing FAEs located north (poleward) of a bright, southward-propagating discrete aurora. The FAEs appeared in multiple regions separated by approximately 500 km longitudinally. The red circle indicates the observing direction of the ASIS spectrograph. (b–d) Watec camera images showing FAEs detected only in the green line, while the discrete aurora was brightest in the red line. (e) Spectrum of the discrete aurora at the magnetic zenith (the red dot in panel a), with strong emissions at 630.0 nm and 636.4 nm.

propagation was faster for the FAEs, and over time, the latitudinal distance between them decreased. Furthermore, as indicated by the orange arrows, the aurora moved eastward on the poleward (north) side and westward on the equatorward (south) side. The shear motion of the aurora near the FAEs was a common feature observed in all events.

## 4 Discussion

In this study, we observed small-scale auroral emissions, namely fragmented auroral-like structures (FAEs) and picket fence structures, over northern Scandinavia on January 1, 2025, using ground-based optical instruments and in-situ data from the Swarm satellite. The key findings are summarized as follows:

- FAEs and picket fence structures were observed near the poleward edge of the auroral oval after a substorm during an intense magnetic storm. They were observed between 20 and 01 MLT.

- The all-sky cameras revealed that FAEs can also appear simultaneously at multiple longitudinally separated locations.

- In addition to the previously known FAEs, we also observed FAEs whose orientations are closely aligned with the local magnetic field in the image plane.

- Thanks to the wide field of view provided by the all-sky cameras, we visualized that the FAEs appeared to follow the motion of red auroras.

Our observations suggest that FAEs are not produced by a single generation mechanism. Among previously reported cases, only our Event 1 showed clearly field-aligned structures; in most cases, FAEs are not aligned with the magnetic field. This implies that the FAEs may be produced by both field-aligned and non-field-aligned acceleration processes of electrons, and they may appear to be aligned only when the non-field-aligned acceleration is weak.

Like FAEs, picket fence structures observed in association with STEVE also exhibit both field-aligned and non-field-aligned components. Semeter et al. (2020) conducted a detailed analysis of the morphology of picket fences observed simultaneously with STEVE and showed that they are not purely field-aligned; instead, small point-like emission sources appear at their lower ends, sometimes forming J-shaped or L-shaped features. In our event, both FAEs and picket fence structures were observed on the same night, which further highlights the similarity between the two phenomena. If this similarity reflects a shared generation process, it may suggest that FAEs, like picket fences, can also consist of both field-aligned and non-field-aligned components.

Although not strongly supported in this case, one possible mechanism for producing non-field-aligned structures is the gradient drift instability (GDI). This idea is motivated by the fact that the FAEs appeared to follow the horizontal motion of red auroras (i.e., motion perpendicular to the field lines) and that there were spatial gradients in electron density in the background F region. GDI grows when the direction of the electron density gradient is parallel to the $\mathbf{E} \times \mathbf{B}$ drift and is known to produce finger-like structures (with horizontal scales of 55–210 km) on the trailing edge of polar cap patches (Hosokawa et al., 2016). This is similar to our observation in which the FAEs appeared on the trailing side of the red aurora in the F region. However,

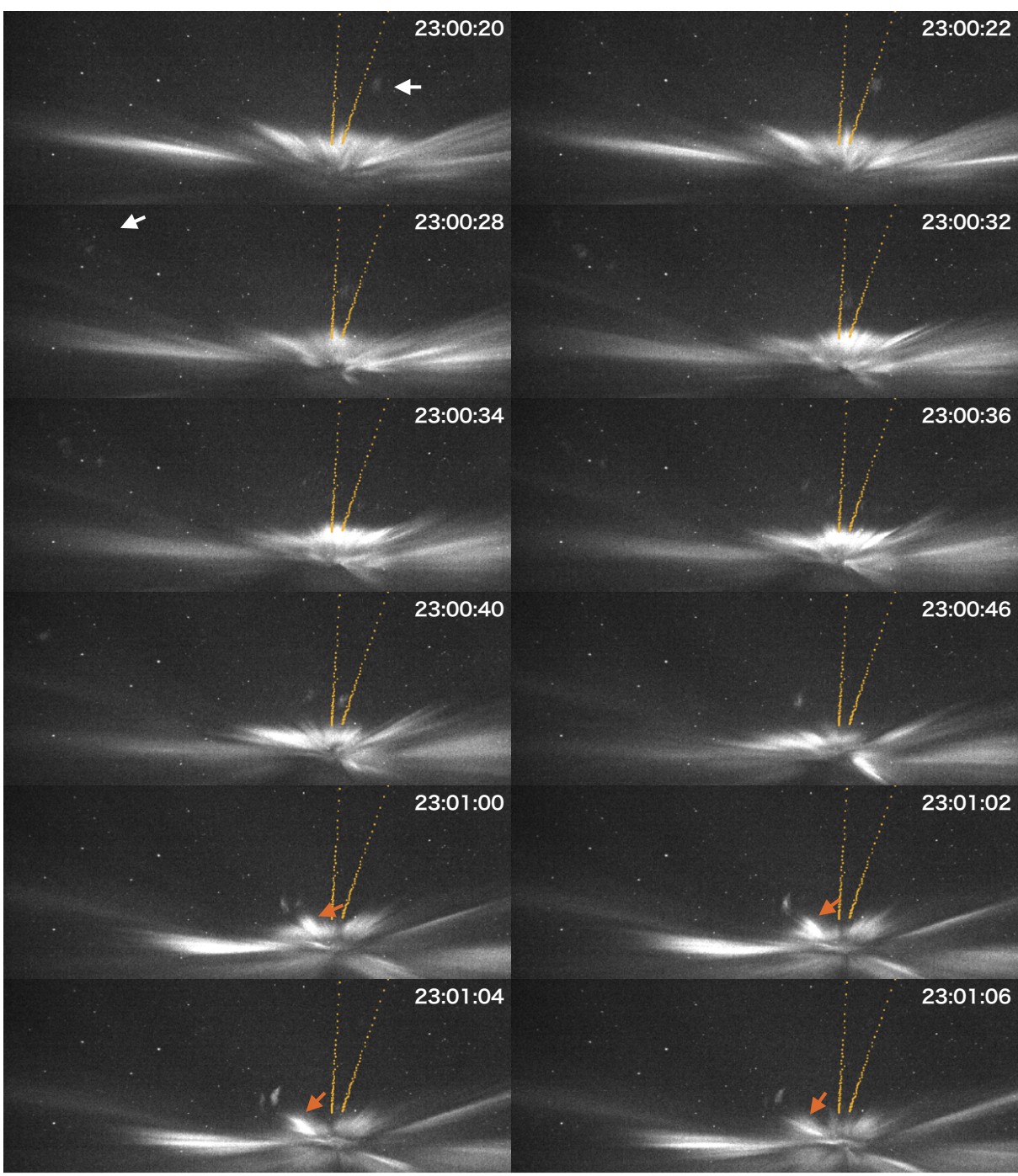

**Figure 12.** qCMOS camera observations of FAEs near the zenith during Event 3. The FAEs appeared around 23:00:20 UT, lacked field-aligned structure, and propagated eastward while the overall distribution shifted southward. The discrete aurora south of the FAEs also moved southward but at a slower speed, reducing the distance between them. Orange arrows indicate shear motion of the aurora, moving eastward on the north side and westward on the south side.

GDI is not considered very effective in the E region, where collisions between plasma and neutral particles are frequent. Explaining green FAEs or picket fence structures, which likely originate in the E region based on their color, with GDI would require resolving this gap in altitude. Nonetheless, since both the FAEs and the picket fence structures appeared to follow the motion of red auroras in the F region, it is still possible that GDI had an indirect influence on their generation.

The morphological feature that most of FAEs do not exhibit field-aligned structures has led to the idea that they may be generated by Farley–Buneman instabilities (FBI), as proposed by Dreyer et al. (2021) and Whiter et al. (2021). FBI occurs when the relative velocity between electrons and ions exceeds the ion acoustic speed, and unlike GDI, it can grow in the E region of the ionosphere. The FAEs observed in this study also showed fine-scale structures on the order of a kilometer and lifetimes shorter than one minute, consistent with characteristics reported in previous studies. Whiter et al. (2021) estimated the $\mathbf{E} \times \mathbf{B}$ drift velocity based on the group and phase velocities of the FAEs and suggested that a strong electric field is required for their generation. Unfortunately, this type of analysis cannot be applied to our event due to the limited temporal resolution of the camera and the lack of radar measurements. Nevertheless, since the aurora displayed rapid and rotation-like motion just before the appearance of the FAEs, the presence of a strong electric field cannot be excluded. Therefore, our observations do not contradict the possibility that the FBI plays an important role in the generation of the FAEs.

In Event 1, the FAEs exhibited field-aligned structures. This raises the possibility that they were generated by precipitating electrons from the magnetosphere. However, no emission from the $N_2^+$ first negative line (with an excitation potential of $\sim 18$ eV) was detected in the Watec ASI data shown in Figure 4, suggesting that this is unlikely. The emissions were detected by the qCMOS camera in Figure 5. Since the filter installed on this camera is designed to cut the forbidden green and red lines and receive prompt emissions from molecular nitrogen and nitrogen ions, the camera would detect $N_2$ first positive emissions (with an excitation potential of $\sim 7$ eV). This indicates that the absence of first negative emissions is not due to a lack of molecular nitrogen in the atmosphere, but rather due to the absence of a mechanism capable of providing the excitation potential required for the first negative band. Therefore, while soft ($< 1$ keV) precipitation to the F region could account for some red emission, the field-aligned FAEs are unlikely to result from precipitation and are instead consistent with local acceleration of electrons in the ionosphere.

When field-aligned FAEs are observed without a signature of electron precipitation from the magnetosphere, one possible generation mechanism of the field-aligned structure is the ionospheric feedback instability (IFI). IFI is triggered by a localized increase in ionospheric conductivity, and numerical simulations have shown that it grows most efficiently at horizontal scales of approximately 1 km or less when the Pedersen conductance is high (Kataoka et al., 2021). This enhances polarization electric fields that oppose the background ionospheric convection electric field, generating upward-propagating Alfvén waves. These waves can excite standing wave modes in the Alfvén resonator. The standing wave modes are sometimes accompanied by field-aligned electric fields, which can accelerate electrons along magnetic field lines. In the present event, the red aurora propagated in a sweeping motion across the sky, and the FAEs appeared in the region where the ionospheric conductivity was likely enhanced. This suggests that Alfvén waves driven by IFI may have contributed to the formation of small-scale and field-aligned auroral structures.

Another possible mechanism for generating field-aligned structures is Ohmic heating caused by strong field-aligned currents (FACs), as discussed by Lanchester et al. (2001) and proposed by Whiter et al. (2021) as a possible generation mechanism for FAEs. In this mechanism, intense FACs occurring in localized regions of the ionosphere can heat electrons. If the electrons gain sufficient excitation energy of ionospheric particles, the resulting emissions may become optically visible. However, while conditions favorable for thermal red-line emission were satisfied (elevated $T_e$ and $N_e$), Figure 6 does not show evidence of strong FACs occurring simultaneously with the appearance of the FAEs. Therefore, this hypothesis was not confirmed from the observations.

We have considered several plasma instabilities that may contribute to the generation of FAEs and picket fence structures. Each instability has different optimal physical parameters for its development, such as ionospheric conductance and collision frequency. These parameters depend on location, i.e., latitude and altitude, and FAEs are more frequently reported in the polar cap, whereas picket fences are commonly observed in the subauroral region. In our study, both phenomena were observed in auroral latitudes, which lies between these two regions, and the field-aligned structure appeared to change over time. These observations suggest that small differences in background plasma parameters may determine whether the resulting structure exhibits a field-aligned morphology. Future statistical studies of the occurrence locations and background conditions of these emissions may help identify the dominant generation mechanisms and clarify which parameters control the development of field-aligned features.

## 5 Conclusions

We conducted a detailed analysis of fragmented auroral-like emissions (FAEs) and picket fence structures observed in northern Scandinavia during a magnetic storm on January 1, 2025, using multiple ground-based optical instruments and the Swarm satellite. This study presents simultaneous observations of both phenomena within the auroral oval, near the poleward edge, expanding their known occurrence beyond the polar cap and subauroral regions. Notably, we found that some FAEs exhibited orientations closely aligned with the local magnetic field in the image plane, which have not been reported previously and may reflect unique generation conditions. These structures tended to appear following the motion of red auroras, suggesting that local electric field structures and enhancements in ionospheric conductivity may be involved in their formation. Possible generation mechanisms include the Farley–Buneman instability and the ionospheric feedback instability, with the latter potentially explaining field-aligned structures through electron acceleration by Alfvén waves. Excitation by precipitating electrons from the magnetosphere is unlikely, as indicated by the absence of emissions from the first negative band of nitrogen molecular ions. This study provides new insights into the observational characteristics and possible generation mechanisms of FAEs and picket fence structures, highlighting the need for further comparative observations and modeling efforts to deepen our understanding.

*Data availability.* The solar wind data from DSCOVR can be downloaded from https://www.ngdc.noaa.gov/dscovr/portal/index.html. The Dst index is provided from https://wdc.kugi.kyoto-u.ac.jp/. The magnetometer data at Kilpisjärvi is available at https://space.fmi.fi/image/

www/index.php? The images from the all-sky color camera can be downloaded from http://darndeb08.cei.uec.ac.jp/~nanjo/public/skibotn_
imgs/2023_season/20240101/. The Watec ASIs data used in this study are publicly available at http://esr.nipr.ac.jp/www/optical/watec/skb/
rawdata2/. The qCMOS camera data are also available at http://gwave.cei.uec.ac.jp/~hosokawa/public/flagments/. The spectral riometer data
are available at http://darndeb08.cei.uec.ac.jp/~nanjo/public/riometer.

*Video supplement.*  Video A1 is available at https://doi.org/10.5446/71371, and Video A2 is available at https://doi.org/10.5446/71372.

## Appendix A:  Supporting figures

### A1    Optical calibration for Watec cameras

Calibration of the Watec camera (model: WAT-910HX/RC) was conducted in February 2018 at the optical calibration facility
at the National Institute of Polar Research (NIPR). The calibration was performed by evaluating the relationship between JPEG
counts (Count$'$) and absolute brightness in Rayleighs. The detailed procedures are described in Ogawa et al. (2020), and the
relationship was approximated by a linear function:

$$\text{Rayleigh} = a \cdot \text{Count}' + b \tag{A1}$$

The calibration yielded the following coefficients:

$$a = 20.947130, \quad b = 79.3542.$$

The average noise level during the calibration was approximately 115 R. The calibration result is illustrated in Figure A1,
which shows the relationship between the corrected count and Rayleigh values.

In addition, flat-field correction was applied to account for vignetting in the all-sky images. The relative sensitivity as a
function of radial distance $r$ (in pixels) from the image center was modeled using the following equation:

$$\frac{\text{Count}(r)}{\text{Count}(0)} = 1 - \alpha \cdot r^2 \tag{A2}$$

The coefficient $\alpha$ was derived from calibration data and found to be $5.81 \times 10^{-6}$. This equation provides a correction of the
radial sensitivity change due to lens characteristics. The calibration data are shown in Figure A2.

These calibration procedures enable quantitative interpretation of auroral brightness at the 428 nm for the all-sky images.
Similar calibration was also conducted to derive the absolute intensity at 558 nm and 630 nm.

### A1    Quantitative comparison between FAE and magnetic field-line inclinations

We quantitatively evaluated the deviation between the observed FAE inclination and the modeled magnetic field–line inclina-
tion on the image. The outline of the analysis is below.

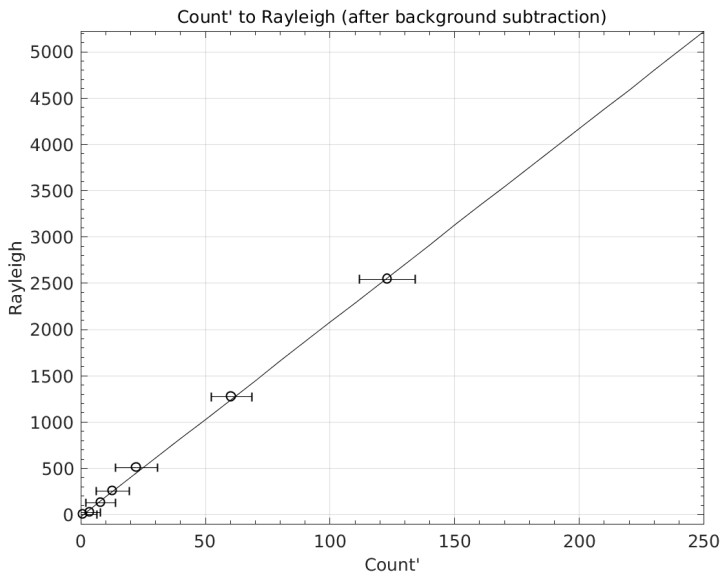

**Figure A1.** Calibrated relationship between gamma-corrected count and Rayleigh for the 428 nm filter.

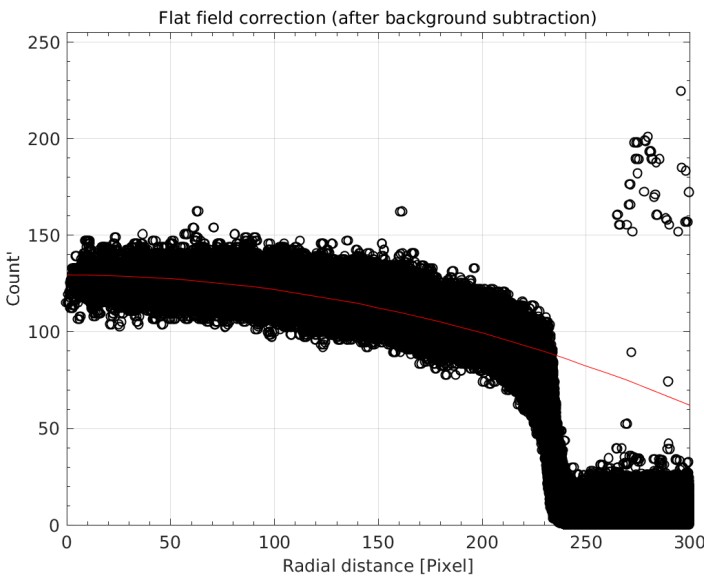

**Figure A2.** Radial sensitivity profile used for flat-field correction.

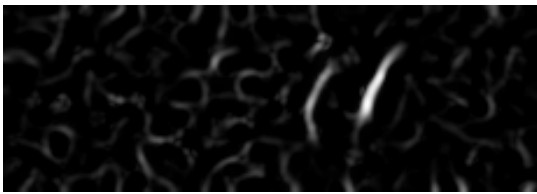

**Figure A3.** Example of the Frangi response image used to enhance faint FAE structures.

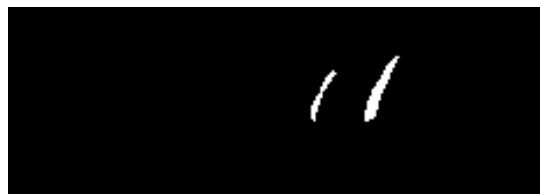

**Figure A4.** Binary mask of the FAE candidates obtained from the Frangi response.

1. Extraction of each FAE

2. Fitting the extracted structure with a line and computing its inclination on the image

3. Computing the model magnetic field inclination in the image and interpolating to all pixels

4. Comparing the inclinations obtained in (2) and (3) and evaluating the mean and scatter of the differences

We processed each image in the interval shown in Figure 5 individually. From each original observational image, we extracted the region of interest (ROI) where the FAE appeared ($x = 450$–$680$ pixels, $y = 160$–$240$ pixels). The FAE observed in this event was faint, with 16-bit pixel values typically ranging from 3150 to 3700. Because this intensity range is close to the background noise level, simple thresholding failed to reliably identify filamentary structures. Therefore, we normalized the ROI using the 5–95 percentile range (to suppress the influence of bright stars), and applied the Frangi vesselness filter, which is designed to enhance elongated structures (Frangi et al., 1998). The resulting Frangi response image is shown in Figure A3.

As seen in Figure A3, two elongated structures are clearly enhanced on the right in the middle of the image. We selected the top $1\%$ of pixels in the Frangi response as initial FAE candidates. Among these, we retained only those pixels whose original intensities lay within 3150–3700, and removed candidate regions with an area smaller than 45 pixels. The resulting binary mask is shown in Figure A4. From these candidate regions, only those with eccentricity greater than 0.8 were considered true FAE structures. In the example shown here, both regions met this criterion.

For each accepted region, we collected all pixel coordinates and computed their centroid. We then applied principal component analysis (PCA) to determine the dominant elongation direction. By projecting each pixel onto the first principal component, we identified the minimum and maximum projection values and defined the corresponding positions along the PCA axis as the endpoints of the fitted line segment. Thus, each FAE was represented by a single straight line defined by these two

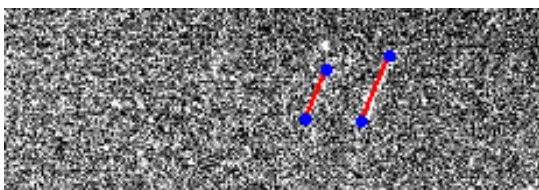

**Figure A5.** Example of line fitting using PCA. The red line shows the fitted segment, and the blue points mark its endpoints.

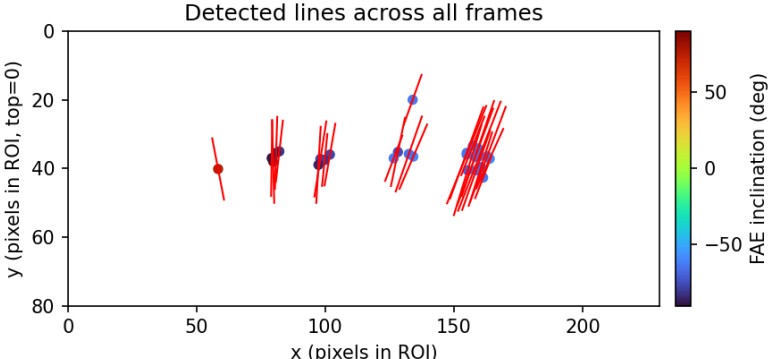

**Figure A6.** Detected FAE line segments across all frames in the ROI. The color of each dot indicates the inclination angle of the corresponding segment.

endpoints. An example result is shown in Figure A5, where the red line indicates the fitted segment and the blue points mark its endpoints.

Although the background image in Figure A5 appears noisy, this is because it was generated only for debugging (scaled to 8-bit using the 5–95 percentile range). The fitting does not use this background image. Repeating this procedure for all frames in the interval produced 26 line segments in total. These segments were plotted together in the ROI coordinate system, with the inclination of each segment indicated by a colored dot placed at its midpoint, as shown in Figure A6.

  We further traced the model magnetic field lines shown as yellow dots in Figure 5 of the main text, using a finer latitude

sampling step of $0.05°$. For each pair of consecutive field–line points, we computed the local inclination in the image using

$$\theta = \arctan 2(\Delta y, \Delta x), \tag{A1}$$

where $\Delta x$ and $\Delta y$ denote the pixel–coordinate differences between two adjacent field–line points. We interpolated these inclination values over the entire ROI using Gaussian filtering so that every pixel has an estimated inclination. The resulting inclination field is shown in Figure A7; the red curves show the model magnetic field lines, and the background color represents

the local inclination.

  With this procedure, we directly compared the inclination of each observed FAE with that of the modeled magnetic field line at the corresponding location. In total, 26 FAE segments were identified from the image sequence, and for each segment

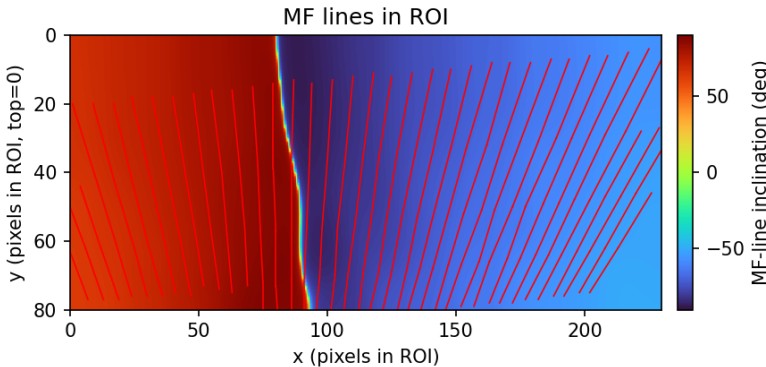

**Figure A7.** Inclination field of the model magnetic field lines in the ROI. Red curves show the traced field lines, and the background color represents the local inclination.

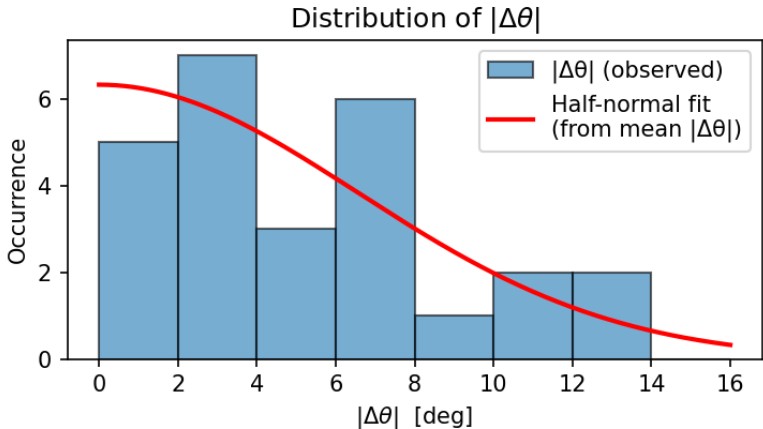

**Figure A8.** Histogram of $|\Delta\theta|$ for the 26 detected FAE segments, together with a half–normal distribution fitted from the observed mean.

we calculated the absolute inclination difference, $|\Delta\theta|$, between the observed structure and the model field-line direction. The resulting distribution yields a mean of $5.23°$ and a standard deviation of $4.01°$, indicating that the measured deviations typically fall within several degrees of the modeled magnetic-field inclination.

The distribution of $|\Delta\theta|$ is shown in the histogram in Figure A8, together with a half–normal probability density function constructed from the observed mean value (the vertical scale is adjusted for visual comparison). Although the number of samples is limited and the histogram does not perfectly follow the analytic curve, the overall characteristics are consistent with a half–normal distribution: nearly half of the samples fall within $0$–$4°$, while the occurrence gradually decreases toward larger values, forming a natural long tail beyond about $8°$.

To interpret the mean and standard deviation, we model the observed inclination difference as

$$\Delta\theta_{\mathrm{obs},i} = \theta_{\mathrm{FAE},i} - \theta_{\mathrm{MF},i} = \Delta\theta_{\mathrm{true}} + e_i, \qquad e_i \sim \mathcal{N}(0,\sigma^2), \tag{A2}$$

where $e_i$ represents the combined uncertainty from the linear fitting of the segments, magnetic field modeling, and geometrical calibration of the observed image. Assuming that the FAE is intrinsically parallel to the true magnetic field, we set $\Delta\theta_{\mathrm{true}} = 0$, giving

$$\Delta\theta_{\mathrm{obs},i} = e_i. \tag{A3}$$

The absolute value $X = |e_i|$ then follows a half–normal distribution with

$$\mathrm{E}[X] = \sigma\sqrt{\frac{2}{\pi}}, \qquad \mathrm{Var}(X) = \sigma^2\left(1 - \frac{2}{\pi}\right). \tag{A4}$$

Using the observed mean value $\overline{|\Delta\theta|} = 5.24°$, we estimate

$$\hat{\sigma} = \overline{|\Delta\theta|}\sqrt{\frac{\pi}{2}} \approx 5.24° \times \sqrt{\frac{\pi}{2}} \approx 6.6°. \tag{A5}$$

Here $\hat{\sigma}$ is the estimated $1\sigma$ uncertainty in $\Delta\theta_{\mathrm{obs},i}$. The predicted standard deviation of $|\Delta\theta|$ is then

$$\sqrt{\mathrm{Var}(|\Delta\theta_{\mathrm{obs}}|)} = \hat{\sigma}\sqrt{1 - \frac{2}{\pi}} \approx 6.6° \times \sqrt{1 - \frac{2}{\pi}} \approx 4.0°, \tag{A6}$$

which agrees very well with the observed value of $4.01°$.

Thus, the observed $|\Delta\theta|$ values are fully consistent with the hypothesis that the FAE is intrinsically parallel to the magnetic field in the image, and that the measured deviation simply reflects the combined $1\sigma$ uncertainty of about 6–7° arising from the linear fitting of segments and geometrical/model errors. Therefore, it is statistically difficult to conclude that "the FAE inclination is systematically different from the model magnetic field line," and the present results are more naturally explained if the FAE orientation is regarded as parallel to the magnetic field in the image within the observational and modeling uncertainties.

*Author contributions.* SN conducted the overall analysis and wrote the first version of the manuscript. SN and MJ operated the observation by the all-sky color camera in Skibotn. AK, SO, and KH operated the observation of the riometer. GC and HL operated the observation of the spectrometer. YO operated the observation by the Watec all-sky camera. KH operated the observation by the qCMOS camera. KH, TS, GC, NP, MJ, SO and MY interpreted and discussed the results with SN. All authors revised the manuscript and approved the final manuscript.

*Competing interests.* At least one of the (co-)authors is a member of the editorial board of Annales Geophysicae.

*Acknowledgements.* We are grateful to the Skibotn Observatory, UiT The Arctic University of Norway, for providing the site for the observations conducted in this study. We thank the Finnish Meteorological Institute and the IMAGE network for providing the magnetometer data

from Kilpisjärvi. We acknowledge the use of the real-time Dst index provided by the World Data Center for Geomagnetism, Kyoto. We also acknowledge the use of data from the Swarm satellites, a mission of the European Space Agency (ESA), and from the DSCOVR satellite, operated by NOAA and NASA. This research has been supported by the Japan Society for the Promotion of Science (grant nos. 21H04518, 21KK0059, 22H00173, 22K21345, 23K22554, and 24H00751) and Norwegian Research Council (contract no. 343302). The first author is a JSPS Overseas Research Fellow.

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
