# Peer review of "Observations of Fragmented Aurora-like Emissions and Picket Fence on the Poleward Edge of the Auroral Oval"

_EGUsphere, 2025_

## Referee Comment (RC1)

The paper by Nanjo et al. reports the first observations of the FAEs and picket fence in auroral latitudes. The authors show that the FAEs are field-aligned and appear simultaneously at multiple locations. The FAEs follow the motion of red auroras. They suggest that the FAEs are caused by local acceleration of electrons. I think the observations in the paper are interesting and worth publication. However, the paper has several major issues, particularly regarding its claim about the first observation, field-aligned nature, Swarm data interpretation, and data availability. I suggest considering the comments described below before recommending publication in ANGEO.

**Major comments**

Line 4, 280, 357. The authors should note that the figures in Partamies et al. (2025) do show green emissions below the continuum emissions, for example in their Figure 1a, 11a, and 12c. Their spectrograph data shows evidence of strong 557.7 nm emissions. While the simultaneous presence of the continuum and picket fence does not appear to be Partamies et al.'s focus, this information, which is available within your author team, should be acknowledged, and the claim of "first observation" should be removed.

Line 175 The authors claim that the picket fence is field-aligned. It appears to be true for some pickets, but a significant part of the picket fence is not field-aligned. See the figure in the PDF version of the comments. The ones marked in red are tilted away from the magnetic field lines. (a) The authors should mention that a significant part of the picket fence is not field-aligned. (b) The magnetic field lines depend on the assumed emission altitude. Please describe the assumption used to draw the magnetic field lines, and discuss uncertainties of the field-alignment considering uncertainties in the altitude.

Line 190-202. Several issues exist regarding the results from Swarm. (a) Swarm did not cross FAE, and therefore the FAC, density and temperature from Swarm cannot be used to discuss the generation conditions of FAE. Swarm provides background plasma conditions at best. This limitation should be mentioned explicitly. (b) Downward region 2 FACs on the red aurora is inconsistent with the converging electric field deduced from Figure 3. The converging electric field should be connected to upward FACs, likely where the region 1 is. (c) Looking at Figure 4a, Swarm crossed the latitude of the FAE before 19:10:30 UT. The FAE location in Figure 6 should be corrected. (d) It is difficult to compare Figure 6 and 4a. Add more tick marks along the satellite trajectory of Figure 4a. Then make the location of the red aurora and FAE in Figure 6 consistent with Figure 4a.

Data availability. Some data in this research are only available upon request, which it does not meet ANGEO's data policy: "Copernicus Publications requests depositing data that correspond to journal articles in reliable (public) data repositories...If the data are not publicly accessible at the time of final publication, the data statement should describe where and when they will appear...Nevertheless, authors should make such embargoed data available to reviewers during the review process in order to foster reproducibility." This data availability issue must be corrected before publication of this paper. https://www.annales-geophysicae.net/policies/data\_policy.html

**Minor comments**

Line 4, 278, 357 Line 4 and 357 state that FAEs are in the oval, but line 278 states the FAEs are poleward of the auroral oval. Please be consistent.

Line 24 Please provide references that show the picket fence is "usually" field-aligned. I'm only aware of the case studies by Semeter et al. If there are no references showing the usual field-aligned occurrence of the picket fence, this sentence should be rewritten to "Semeter et al. suggested that the picket fence is field-aligned."

Line 38 Nanjo et al. (2024) did not demonstrate that the emission similar to STEVE is aurora. "Aurora" should be removed.

Section 2 should provide references to each of the instruments, unless this is the first paper that uses the data from the instruments.

Line 137 Auroral explosion is not a widely used term in auroral physics. Change this to "poleward expansion."

Figure 3 and 4. It is unclear why Figure 3 presents many images without FAE. FAE is shown in only one image in Figure 4 with a gap in time from Figure 3. Please show more images between 19:08:33 and 19:11:06 UT, and describe how the red aurora changes during the FAE appearance.

Line 221 Describe what assumptions were used to determine the latitude.

Line 278. Change "poleward side of" to "near the poleward edge of." The poleward side means near but poleward of the auroral oval, but it is not what the authors say in the conclusion.

---

## Referee Comment (RC3)

I appreciate the authors for considering my original comments and explaining proposed revisions. The proposed revisions for the Swarm data interpretation, data availability, and the minor comments seem appropriate. However, I'm not convinced by the responses about the comments on the prior work and the field-aligned structures. My follow-up comments are described below:

(1) > all events discussed by Partamies et al. (2025) are dayside cases observed at Svalbard

This is not entirely true because their event in Figure 14c is on the nightside (19 MLT). Figure 14c shows picket fence-like emissions.

I'm not sure why Svalbard vs. Scandinavia can claim the uniqueness of the paper. Both papers show observations near the poleward edge of the auroral oval, providing a similar geophysical context. The latitude is different, but the oval location changes with magnetic activity.

Considering these, I still urge that the observations by Partamies et al. (2025) should be introduced and acknowledged. Although the point of your paper is the nightside observations, the common nature of the emissions near the poleward edge of the oval should be stated clearly.

(2) > In contrast, for Event 1, we consider that the structures are essentially field-aligned. We did not extract only those FAEs that happened to align within randomly extended structures; rather, the sequence shows orientations parallel to the local field in general... As you noted, some FAEs in individual snapshots do not appear perfectly parallel to the model field lines. However, when following the sequence, as the FAEs propagate from west to east, their orientation tracks the change in the model field-line inclination across the FoV

I don't think that revised Figure 5 demonstrates this point. I have copied my markups to revised Figure 5 (see the PDF version of the comment). My previous comment pointed out that a significant part of the picket fence is not field aligned, and this version of the figure still shows the same issue. Although I agree that the orientation tracks the changes of field line orientation overall, the misalignment of a significant part of the picket fence does not support the conclusion that the picket fence for Event 1 is field-aligned.

---

## Author Comment (AC1)

We deeply thank the reviewer for the careful and constructive assessment of our manuscript. We are also grateful for the rapid review, which enabled us to improve the paper efficiently. Below are our point-by-point responses to the comments and questions. Since this response includes figures, we would appreciate it if you could read the PDF version attached.

> Line 4, 280, 357. The authors should note that the figures in Partamies et al. (2025) do show green emissions below the continuum emissions, for example in their Figure 1a, 11a, and 12c. Their spectrograph data shows evidence of strong 557.7 nm emissions. While the simultaneous presence of the continuum and picket fence does not appear to be Partamies et al.'s focus, this information, which is available within your author team, should be acknowledged, and the claim of "first observation" should be removed.

Thank you for pointing out our insufficient explanation. We will explicitly acknowledge that Partamies et al. (2025) reported continuum emissions together with co-located green emissions in the introduction. However, all events discussed by Partamies et al. (2025) are dayside cases observed at Svalbard, whereas our study analyzes nightside events observed in Scandinavia, so the local-time context differs. Because this MLT difference may be useful for future discussion of generation mechanisms, we will make the distinction explicit by adding the term "nightside" throughout the manuscript. In addition, the statement at Line 280 concerns a feature not reported by Partamies et al. (2025)—namely, the simultaneous occurrence of FAEs at multiple longitudinally separated locations in auroral latitudes—and we would like to keep this statement.

> Line 175 The authors claim that the picket fence is field-aligned. It appears to be true for some pickets, but a significant part of the picket fence is not field-aligned. See the figure in the PDF version of the comments. The ones marked in red are tilted away from the magnetic field lines. (a) The authors should mention that a significant part of the picket fence is not field-aligned. (b) The magnetic field lines depend on the assumed emission altitude. Please describe the assumption used to draw the magnetic field lines, and discuss uncertainties of the field-alignment considering uncertainties in the altitude.

Thank you for the helpful comment, and we apologize if our previous figure caused confusion. Our manuscript reports three events (on the same night) separately. For Events 2 and 3, we do not find any field-aligned FAEs. In contrast, for Event 1, we consider that the structures are essentially field-aligned. We did not extract only those FAEs that happened to align within randomly extended structures; rather, the sequence shows orientations parallel to the local field in general.

To clarify this point, we provide a revised Figure 5 below and will state the assumptions to calculate the field lines: the lowest points (footpoints) of the field lines are placed at geographic latitude 69.6°, altitude 110 km, and geographic longitudes 19.0°–21.0° in 0.2° steps. As you noted, some FAEs in individual snapshots do not appear perfectly

parallel to the model field lines. However, when following the sequence, as the FAEs propagate from west to east, their orientation tracks the change in the model field-line inclination across the FoV (features that are oblique on the western side become more nearly vertical where the local field-line direction is more vertical). Such a trend is not consistent with mechanisms that are independent of the magnetic-field direction. While some snapshots indeed show FAEs not perfectly parallel to the model field lines, the systematic alignment supports the view that a field-aligned mechanism essentially contributes to their formation. Accordingly, we wish to avoid the characterization that a 'significant part' of the FAEs is non-field-aligned.

Revised Figure 5

> Line 190-202. Several issues exist regarding the results from Swarm. (a) Swarm did not cross FAE, and therefore the FAC, density and temperature from Swarm cannot be used to discuss the generation conditions of FAE. Swarm provides background plasma conditions at best. This limitation should be mentioned explicitly. (b) Downward region 2 FACs on the red aurora is inconsistent with the converging electric field deduced from Figure 3. The converging electric field should be connected to upward FACs, likely where the region 1 is. (c) Looking at Figure 4a, Swarm crossed the latitude of the FAE before 19:10:30 UT. The FAE location in Figure 6 should be corrected. (d) It is difficult to compare Figure 6 and 4a. Add more tick marks along the satellite trajectory of Figure 4a. Then make the location of the red aurora and FAE in Figure 6 consistent with Figure 4a.

Thank you very much for these helpful suggestions. We have revised Figure 4a and Figure 6 and reinterpreted the Swarm data accordingly, as summarized below.

**(a) Scope of Swarm measurements for this study.**

We agree that FAEs occurred in a very confined region and that Swarm did not cross them directly. We will state explicitly that Swarm's measurements in this case are used only as background conditions, not to infer FAE generation directly.

**(b) FAC sense over the red aurora.**

As seen in the revised Figure 4a, the red aurora (marked by red arrows) extends longitudinally and overlaps the Swarm tracks. Swarm A and C appear to cross the red aurora near 19:11:15 UT and 19:11:00 UT, respectively. In the revised Figure 6, this interval would correspond to the electron-density enhancement labeled "Red aurora?". In our first preprint this was labeled "FAEs?", but we now withdraw that labeling and interpret the enhancement as a crossing of the red aurora. This region corresponds to upward Region 1 FACs (see Figure 6b), addressing the inconsistency you noted.

**(c) FAE latitude vs. Swarm timing.**

The revised Figure 4a shows that Swarm passed the latitude where FAEs later appeared at about 19:10:30 UT. However, as you correctly point out, the FAEs were spatially offset from the Swarm track. We therefore avoid using Swarm measurements to deduce FAE properties in the formal revision.

**(d) Figure labels.**

To improve comparability, Figure 4a now includes diamonds every 30 s along the Swarm trajectories, and Figure 6 has matching vertical grid intervals (every 30 s). In addition to the figures, the main text will be revised to ensure consistency with these interpretations. We are grateful for your comments; they helped us correct our interpretation and clarify the manuscript.

Revised Figure 4

Revised Figure 6

**Data availability:**

The Watec ASIs data used in this study are publicly available at <a href="http://esr.nipr.ac.jp/www/optical/watec/skb/rawdata2/">http://esr.nipr.ac.jp/www/optical/watec/skb/rawdata2/</a>. The qCMOS camera data are also available at <a href="http://gwave.cei.uec.ac.jp/~hosokawa/public/flagments/">http://gwave.cei.uec.ac.jp/~hosokawa/public/flagments/</a>. The ASIS data are available at: <a href="https://asis.aeronomie.be/papers">https://asis.aeronomie.be/papers</a>. The spectral riometer data are currently being processed for public release, and will be made available in the formal revision of the manuscript.

**Minor comments:**

> Line 4, 278, 357 Line 4 and 357 state that FAEs are in the oval, but line 278 states the FAEs are poleward of the auroral oval. Please be consistent.

We will revise Lines 4, 278, and 357 to consistently describe the FAE region as near the poleward edge of the auroral oval.

> Line 24 Please provide references that show the picket fence is "usually" field-aligned. I'm only aware of the case studies by Semeter et al. If there are no references showing the usual field-aligned occurrence of the picket fence, this sentence should be rewritten to "Semeter et al. suggested that the picket fence is field-aligned."

We will revise it as suggested.

> Line 38 Nanjo et al. (2024) did not demonstrate that the emission similar to STEVE is aurora. "Aurora" should be removed.

We will revise "auroral emissions" to "emissions" as suggested.

> Section 2 should provide references to each of the instruments, unless this is the first paper that uses the data from the instruments.

Thank you for your suggestion.

**Ground-based magnetometer (IMAGE)**: This is part of the IMAGE magnetometer network. We will add the reference in the formal revision (10.1029/2008JA013682).

ASIS and qCMOS camera: These instruments were not developed exclusively for this study, but they have only recently become operational, and a dedicated instrument paper has not yet been published. We believe the technical specifications currently described are sufficient for interpreting the data used here.

> Line 137 Auroral explosion is not a widely used term in auroral physics. Change this to "poleward expansion."

We will revise it as suggested.

> Figure 3 and 4. It is unclear why Figure 3 presents many images without FAE. FAE is shown in only one image in Figure 4 with a gap in time from Figure 3. Please show more images between 19:08:33 and 19:11:06 UT, and describe how the red aurora changes during the FAE appearance.

Thank you for the comment. In Figure 3, we show multiple frames to illustrate the motion of the red aurora, as indicated by the white arrows. The frames between 19:08:33 UT and 19:11:06 UT that connect Figure 3 and Figure 4 are already provided

as Video A1 (Supplement). As can be seen there, the red aurora exhibits no pronounced evolution during this interval; therefore, adding more still images would not materially improve clarity. We will add a brief sentence in the text describing this point.

> Line 221 Describe what assumptions were used to determine the latitude.

We did not determine the latitude by assumption. Rather, we chose the latitude slice for which the features captured from Abisko and Skibotn align most closely in projection, minimizing their separation in the longitude-altitude plane. We will explain this in the text.

> Line 278. Change "poleward side of" to "near the poleward edge of." The poleward side means near but poleward of the auroral oval, but it is not what the authors say in the conclusion.

We will revise it as suggested.

---

## Author Comment (AC2)

Thank you for the careful and constructive assessment, and for the quick turnaround. Below are our point-by-point responses to your comments and questions. As this reply includes figures, please refer to the attached PDF for proper viewing.

> My main additional question and concern are about the method used to determine whether the FAEs are field-aligned. The alignment of FAEs in space is three-dimensional, whereas both the camera view and the T89 model mapping are two-dimensional. Even if the field-line mapping appears aligned with the FAEs in a 2D projection, this does not necessarily mean the FAEs are actually parallel to the field lines in 3D space, given the lack of information about the third dimension. This method can demonstrate that the FAEs are not field-aligned in Events 2 and 3, but it seems insufficient to conclude that the FAE structures are field-aligned in Event 1. In this case, it might be premature to draw the conclusion stated in Lines 358–359.

Thank you for this important comment. This issue was partially raised by Referee #1 as well, so some of our response below necessarily overlaps.

We agree with your concern: model magnetic field lines are defined in three dimensions, whereas images by a camera are two-dimensional. Any projection of aurora from one site therefore requires an assumed emission height.

That said, the relevant heights are constrained. The green emissions typically peak near 110 km (e.g., Whiter et al., 2023). Dreyer et al. (2021) also inferred that FAEs occur at altitudes around 110 km. Also in our data, green emissions with morphology similar to Event 1 occur within 110–140 km (see Figure 8). Based on this, we drew model field lines started at altitude 110 km, geographic latitude 69.9°, and longitudes 19.0°–21.0° at 0.2° steps. As shown in the figure below, these field lines converge slightly below the FoV center (the magnetic zenith), and the FAEs in panels (b–i) appear approximately parallel to the modeled local field-lines and would converge toward a similar point. Importantly, when we vary the starting height to 100 km or 140 km, the magnetic-zenith location is essentially unchanged, and the alignment of the FAEs with the local field lines remains the same. In other words, the conclusion that the FAEs tend to follow the local field-line direction holds across a reasonable range of altitudes.

We are not identifying strict 3D distributions of emissions. Our point is that the FAEs are consistent with alignment to the local modeled field lines within a reasonable emission-height range. We will state this limitation in the main text in the formal review.

**Minor issue:**

We will also correct DOI links. Thank you very much for checking.

**Reference:**

Whiter, D. K. et al.: The altitude of green OI 557.7 nm and blue N2+ 427.8 nm aurora, Ann. Geophys., 41, 1–12, https://doi.org/10.5194/angeo-41-1-2023, 2023.

Dreyer, J. et al.: Characteristics of fragmented aurora-like emissions (FAEs) observed on Svalbard, Ann. Geophys., 39, 277–288, https://doi.org/10.5194/angeo-39-277-2021, 2021

Revised Figure 5

---

## Author Comment (AC3)

Thank you very much again for your quick and thoughtful feedback on our response. Since this response includes several figures, we would appreciate it if you could read the attached PDF version.

>> all events discussed by Partamies et al. (2025) are dayside cases observed at Svalbard

This is not entirely true because their event in Figure 14c is on the nightside (19 MLT). Figure 14c shows picket fence-like emissions. I'm not sure why Svalbard vs. Scandinavia can claim the uniqueness of the paper. Both papers show observations near the poleward edge of the auroral oval, providing a similar geophysical context. The latitude is different, but the oval location changes with magnetic activity. Considering these, I still urge that the observations by Partamies et al. (2025) should be introduced and acknowledged. Although the point of your paper is the nightside observations, the common nature of the emissions near the poleward edge of the oval should be stated clearly.

Thank you for pointing this out. You are right that Partamies et al. (2025) present an event around 19 MLT. We had overlooked this case and therefore incorrectly summarized their study as dealing only with dayside events from Svalbard. We will correct this in the revised manuscript.

We also agree that, once the shift of the auroral oval with geomagnetic activity is taken into account, the distinction between Svalbard and northern Scandinavia alone does not by itself provide scientific uniqueness. Partamies et al. (2025) is an important prior work for our study, and, as you point out, some of their findings overlap with our results in that they show similar emissions near the poleward edge of the auroral oval.

At the same time, our observations are not entirely overlapping with theirs. For example, by using a high-spatiotemporal-resolution camera, we are able to resolve additional morphological characteristics of the structures (e.g., their small-scale spacing, temporal evolution, and relation to the background auroras) that were not analyzed in detail in Partamies et al. (2025), since these features were not the main focus of their study. These differences are not intended to suggest any inconsistency or error in Partamies et al. (2025); rather, we see our analysis as complementary to theirs, adding higher-resolution morphological details of green emissions. In the revised manuscript, we will introduce and acknowledge their results more clearly, and explicitly state both the common nature of the emissions occurring near the poleward edge of the oval and how our observations extend their work.

> I don't think that revised Figure 5 demonstrates this point. I have copied my markups to revised Figure 5 (see the PDF version of the comment). My previous comment pointed out that a significant part of the picket fence is not field aligned, and this version of the figure still shows the same issue. Although I agree that the orientation tracks the changes of field line orientation overall, the misalignment of a significant part of the picket fence does not support the conclusion that the picket fence for Event 1 is field-aligned.

Thank you for your comment. Following your suggestion, we attempted to quantitatively evaluate the deviation between the observed FAE inclination and the modeled magnetic field-line inclination. The detailed analysis procedure will be included in the main manuscript as an appendix in the formal revision.

As a result, we were able to automatically extract 26 FAE structures, and the absolute difference between their inclinations and those of the modeled magnetic field lines was found to be 5.23° in mean and 4.01° in standard deviation.

As discussed below, these values naturally arise when we consider the combined uncertainties associated with the linear fitting of the FAE, possible errors in the magnetic field model itself, and the uncertainty in the qCMOS camera's geometrical calibration. Therefore, it is statistically difficult to conclude that "the FAE inclination is systematically different from the inclination of the model magnetic field line." As we noted in our previous response, although the spatial alignment between the two is not perfect, we do not have sufficient grounds to claim a significant departure of FAE inclination from the model field-line direction. Rather, we believe that the FAE orientation is fundamentally aligned with the magnetic field, consistent with our previous reply. Nevertheless, to avoid emphasizing perfect agreement (i.e., that both angles should be exactly 0° everywhere), we will revise the text to make the interpretation clearer and more transparent.

**## Outline of the analysis**

- (1) Extraction of each FAE
- (2) Fitting the extracted structure with a line and computing its inclination on the image
- (3) Computing the model magnetic field inclination in the image and interpolating to all pixels
- (4) Comparing the inclinations obtained in (2) and (3) and evaluating the mean and scatter of the differences

(1)

We processed each image in the interval shown in Figure 5 individually. From each original observational image, we extracted the region of interest (ROI) where the FAE appeared (x = 450-680 pix, y = 160-240 pix). The FAE observed in this event was faint, with 16-bit pixel values typically ranging from 3150 to 3700. Because this intensity range is close to the background noise level, simple thresholding failed to reliably identify filamentary structures. Therefore, we normalized the ROI using the 5–95 percentile range (to suppress the influence of bright stars), and applied the Frangi vesselness filter, which is designed to enhance elongated structures such as blood vessels (Frangi et al., 1998). The resulting Frangi response image applied to Figure 5d is shown below.

As seen in the figure, two elongated structures are clearly enhanced on the right in the middle of the image. We selected the top 1% of pixels in the Frangi response as initial FAE candidates. Among these, we retained only those pixels whose original intensities lay within 3150–3700, and removed candidate regions with an area smaller than 45 pixels. The resulting binary mask is shown below.

From these candidate regions, only those with eccentricity greater than 0.8 were considered true FAE structures. In the example shown, both regions met this criterion.

(2)

For each accepted region, we collected all pixel coordinates and computed their centroid. We then applied principal component analysis (PCA) to determine the dominant elongation direction. By projecting each pixel onto the first principal component, we identified the minimum and maximum projection values, and defined the corresponding positions along the PCA axis as the endpoints of the fitted line segment. Thus, each FAE was represented by a single straight line defined by these two endpoints. An example result is shown below, where the red line indicates the fitted segment and the blue points mark its endpoints.

Although the background image appears noisy, this is because it was generated only for debugging (scaled to 8-bit using the 5–95 percentile range). The fitting in step (2) does not use this background image.

Repeating this procedure for all frames in the interval produced 26 line segments in total. These segments were plotted together in the ROI coordinate system, with the inclination of each segment indicated by a colored dot placed at its midpoint (figure below).

We further traced the magnetic field lines shown as yellow dots in revised Figure 5, using a finer latitude sampling (0.05° step). For each pair of consecutive field-line points, we computed the local inclination in the image using  $\theta = \arctan 2(\Delta y, \Delta x)$ , where  $\Delta x$  and  $\Delta y$  denote the pixel-coordinate differences between two adjacent field-line points. We interpolated these inclination values over the entire ROI using Gaussian filtering so that every pixel has an estimated inclination. The resulting inclination field is shown below; the red curves show the model magnetic field lines and the background color represents the local inclination.

With this, we were able to directly compare the observed FAE inclination with the modeled field-line inclination at the same locations.

(4) For the 26 detected structures, the absolute difference  $|\Delta\theta|$  between the observed FAE inclination and the modeled magnetic field-line inclination was computed. The results are:

Number of extracted segments: 26

Mean  $|\Delta\theta|$ : 5.23 (deg)

Standard deviation  $|\Delta\theta|$ : 4.01 (deg)

The distribution of  $|\Delta\theta|$  is shown in the histogram below. For reference, we superimposed the half-normal probability density function derived from the observed mean value (the vertical scale is multiplied by a constant for clarity). Although the sample size is small and the histogram does not perfectly match the curve, the following qualitative features are consistent:

- (a) Many samples (12; 46% of all) lie within 0-4°, and
- (b) the frequency decreases for values above 8°, producing a natural long tail.

Thus, the observed distribution is broadly consistent with a half-normal distribution.

To interpret the mean and standard deviation, we model the observed inclination difference as

$$\Delta \theta_{\text{obs},i} = \theta_{\text{FAE},i} - \theta_{\text{MF},i} = \Delta \theta_{\text{true}} + \varepsilon_i, \qquad \varepsilon_i \sim \mathcal{N}(0,\sigma^2),$$

where  $\varepsilon_i$  represents the combined uncertainty from the linear fitting of the segments, magnetic field-line modeling, and geometrical calibration of the observed image. Assuming that the FAE is intrinsically parallel to the true magnetic field in the image, we set  $\Delta\theta_{\rm true}=0$ , giving

$$\Delta \theta_{\text{obs}\,i} = \varepsilon_i$$
.

The absolute value  $X = |\varepsilon_i|$  then follows a half-normal distribution with

$$\mathbb{E}[X] = \sigma \sqrt{\frac{2}{\pi}}, \quad \operatorname{Var}(X) = \sigma^2 \left(1 - \frac{2}{\pi}\right).$$

Using the observed mean value  $\overline{|\Delta\theta|} = 5.24^{\circ}$ , we estimate  $1\sigma$  uncertainty in  $\Delta\theta_{{\rm obs},i}$  as

$$\hat{\sigma} = \overline{|\Delta\theta|} \sqrt{\frac{\pi}{2}} \approx 5.24^{\circ} \times \sqrt{\frac{\pi}{2}} \approx 6.6^{\circ}.$$

The predicted standard deviation of  $|\Delta\theta|$  is then

$$\sqrt{\text{Var}(|\Delta\theta_{\text{obs}}|)} = \hat{\sigma}\sqrt{1 - \frac{2}{\pi}} \approx 6.6^{\circ} \times \sqrt{1 - \frac{2}{\pi}} \approx 4.0^{\circ},$$

which agrees very well with the observed value of 4.01°.

Thus, the observed  $|\Delta\theta|$  values are fully consistent with the hypothesis that the FAE is intrinsically parallel to the magnetic field in the image, and that the measured deviation simply reflects the combined  $1\sigma$  uncertainty of 6–7° arising from the linear fitting of segments and geometrical/model errors. Therefore, it is statistically difficult to conclude that "the FAE inclination is systematically different from the model magnetic field line," and the present results are more naturally explained if the FAE orientation is regarded as parallel to the magnetic field within the observational and modeling uncertainties.

**Reference:**

Frangi, A.F., Niessen, W.J., Vincken, K.L., Viergever, M.A. (1998). Multiscale vessel enhancement filtering. In: Wells, W.M., Colchester, A., Delp, S. (eds) Medical Image Computing and Computer-Assisted Intervention — MICCAI'98. MICCAI 1998. Lecture Notes in Computer Science, vol 1496. Springer, Berlin, Heidelberg. https://doi.org/10.1007/BFb0056195